# Genotyping-by-sequencing-based identification of *Arabidopsis* pattern recognition receptor RLP32 recognizing proteobacterial translation initiation factor IF1

Li Fan[1,9], Katja Fröhlich [1,9], Eric Melzer[1,2,9], Rory N. Pruitt[1], Isabell Albert [1,3], Lisha Zhang [1], Anna Joe[1], Chenlei Hua[1], Yanyue Song[1], Markus Albert [1,3], Sang-Tae Kim [4,5], Detlef Weigel[4], Cyril Zipfel [6], Eunyoung Chae [4,7✉], Andrea A. Gust [1✉] & Thorsten Nürnberger [1,8✉]

Activation of plant pattern-triggered immunity (PTI) relies on the recognition of microbe-derived structures, termed patterns, through plant-encoded surface-resident pattern recognition receptors (PRRs). We show that proteobacterial translation initiation factor 1 (IF1) triggers PTI in *Arabidopsis thaliana* and related Brassicaceae species. Unlike for most other immunogenic patterns, IF1 elicitor activity cannot be assigned to a small peptide epitope, suggesting that tertiary fold features are required for IF1 receptor activation. We have deployed natural variation in IF1 sensitivity to identify *Arabidopsis* leucine-rich repeat (LRR) receptor-like protein 32 (RLP32) as IF1 receptor using a restriction site-associated DNA sequencing approach. RLP32 confers IF1 sensitivity to *rlp32* mutants, IF1-insensitive *Arabidopsis* accessions and IF1-insensitive *Nicotiana benthamiana*, binds IF1 specifically and forms complexes with LRR receptor kinases SOBIR1 and BAK1 to mediate signaling. Similar to other PRRs, RLP32 confers resistance to *Pseudomonas syringae*, highlighting an unexpectedly complex array of bacterial pattern sensors within a single plant species.

[1] Center of Plant Molecular Biology (ZMBP), University of Tübingen, Tübingen, Germany. [2] BioChem agrar, Labor für biologische und chemische Analytik GmbH, Machern, Germany. [3] Department of Biology, University of Erlangen-Nürnberg, Erlangen, Germany. [4] Department of Molecular Biology, Max Planck Institute for Developmental Biology, Tübingen, Germany. [5] Department of Medical & Biological Sciences, The Catholic University of Korea, Bucheon-si, South Korea. [6] Institute of Plant and Microbial Biology, Zürich-Basel Plant Science Center, University of Zürich, Zürich, Switzerland. [7] Department of Biological Sciences, National University of Singapore, Singapore, Singapore. [8] Department of Biochemistry, University of Johannesburg, Johannesburg, South Africa. [9] These authors contributed equally: Li Fan, Katja Fröhlich, Eric Melzer. ✉email: dbsce@nus.edu.sg; andrea.gust@zmbp.uni-tuebingen.de; nuernberger@uni-tuebingen.de

Metazoans and plants employ innate immune systems to cope with microbial infections. Immunogenic microbe-derived signatures from pathogenic, commensal, or beneficial microbes, collectively referred to as microbe or pathogen-associated molecular patterns (MAMPs/PAMPs), serve as ligands for host-encoded cell-surface pattern-recognition receptors (PRRs)[1,2]. Pattern recognition and subsequent initiation of intracellular immune signaling culminates in the activation of antimicrobial defenses, ultimately restricting pathogen spread.

In plants, pattern-triggered immunity (PTI) controls attempted infections by host nonadapted microbes and contributes to basal immunity against host-adapted pathogens[1–4]. PTI suppression by pathogen-derived effectors is an element of successful infection of host plants by host-adapted microbes. Effector-mediated host susceptibility has driven the evolution of intracellular immune receptors that recognize effector activities on host-plant targets and mediate activation of immunity to host-adapted pathogens, a process termed effector-triggered immunity (ETI)[5,6]. Mutual potentiation of PTI and ETI pathways in *Arabidopsis thaliana* (hereafter *Arabidopsis*) has been proposed, suggesting mechanistic links between these two layers of plant immunity[7,8]. An *Arabidopsis* plasma membrane-associated intracellular signaling complex linking helper NLRs (NUCLEOTIDE-BINDING LEUCINE-RICH REPEAT RECEPTORS) from the ADR1 (ACTIVATED DISEASE RESISTANCE 1) family and the lipase-like proteins EDS1 (ENHANCED DISEASE SUSCEPTIBILITY 1) and PAD4 (PHYTOALEXIN DEFICIENT 4) to plant PRRs may provide a convergence point for PTI and ETI signaling[9].

Plant cell-surface-resident PRRs are distinguished by structurally diverse extracellular domains for ligand binding, including leucine-rich repeat (LRR), lysin motif (LysM), or lectin domains[1,2]. LRR-domain proteins predominantly mediate the perception of microbe-derived proteins or peptides[10], and are classified as either LRR receptor kinases (LRR-RKs) or LRR-receptor proteins (LRR-RPs), depending on the presence or absence of a cytoplasmic kinase domain[1,2]. LRR-RPs form constitutive heteromeric complexes with the adaptor kinase SUPPRESSOR OF BRASSINOSTEROID INSENSITIVE 1 (BRI1)-ASSOCIATED KINASE (BAK1)-INTERACTING RECEPTOR KINASE 1 (SOBIR1) and, like LRR-RKs, bind to members of the SOMATIC EMBRYOGENESIS RECEPTOR KINASE (SERK) protein family in a ligand-dependent fashion[1,11]. PRR complex formation subsequently triggers downstream signaling pathways which, though overlapping, differ, depending on the receptor[12].

In the past two decades, several plant PRRs recognizing molecularly defined patterns from bacteria, fungi, and oomycetes have been identified[1,2,4]. In addition, immune-stimulating insect or parasitic plant-derived patterns and their cognate immune sensors have been elucidated[13–15]. *Arabidopsis* LRR-RK FLAGELLIN SENSING 2 (FLS2) recognizes fragments of bacterial flagellins containing a 22-amino-acid motif (flg22)[16]. Other *Arabidopsis* LRR-type sensors for bacteria-derived patterns include ELONGATION FACTOR THERMO-UNSTABLE RECEPTOR (EFR), XANTHINE/PERMEASE SENSING 1 (XPS1), and RLP23[17–19]. Similar to the FLS2 ligand flg22, small immunogenic epitopes within these patterns have been defined, including elf18 (from ELONGATION FACTOR THERMO-UNSTABLE), xup25 (from XANTHINE PERMEASE), and nlp20 (from NECROSIS AND ETHYLENE-INDUCING PROTEIN 1-LIKE PROTEINs)[16,18,19]. FLS2 activities have been found throughout higher plants, which is in contrast to the majority of PRRs, which are often restricted to individual taxa[1]. These include CSPR (CSP22 RESPONSIVENESS) and CORE (COLD SHOCK PROTEIN RECEPTOR), Solanaceae receptors for bacterial cold-shock protein fragment csp22, and tomato FLS3,

which recognizes a flagellin fragment (flgII-28) unrelated to flg22[20–22].

Accumulating evidence suggests that plants employ multiple PRRs to sense a given microbe[1]. For example, in addition to two immune sensors recognizing bacterial medium-chain 3-hydroxy fatty acids and peptidoglycans[23,24], *Arabidopsis* uses the FLS2, EFR, and XPS1 receptors to recognize three proteinaceous *Pseudomonas syringae*-derived peptide patterns[16,18,19]. This complexity is likely to increase, given that the *Arabidopsis* genome encodes more than 600 transmembrane-receptor-like proteins[25–27]. Besides an evident academic interest that drives the identification of plant PRRs and their microbe-derived patterns, such receptors may be exploited in breeding for engineering durable disease resistance in crop plants. Interfamily transfer of plant PRRs into crops has been demonstrated to confer novel pattern-recognition capabilities and enhanced immunity to infection by host-adapted pathogens[28]. Thus, PRR combinations employed in transgenic crops may become an important tool to reduce crop losses and to secure global food security.

Here, we report the identification of *Arabidopsis* RLP32 as a sensor of proteobacterial protein translation-initiation factor 1 (IF1). Different from other PRRs, RLP32 activation requires the complete IF1 molecule, suggesting that RLP32 senses the IF1 tertiary fold. Our findings indicate that bacterial pattern-recognition systems in plants make use of a wide diversity of biochemical principles and that *Arabidopsis* employs numerous receptor systems to sense *P. syringae*.

## Results

**Ralstonia solanacearum-derived pattern recognition in Arabidopsis.** The plant pathogen *R. solanacearum* has previously been reported to produce *Arabidopsis* defense elicitors other than bacterial flagellin[29]. In agreement with this, we found that protein fractions from *R. solanacearum* grown in liquid culture elicited plant defenses in the *Arabidopsis fls2 efr* double mutant (Supplementary Fig. 1), suggesting that *R. solanacearum* elicitor activity (RsE) is not only different from flagellin-derived flg22, but also from elf18. RsE induces the production of reactive oxygen species (ROS), callose and the plant hormone ethylene, the phosphorylation of mitogen-activated protein kinases MPK3 and MPK6, as well as enhanced expression of the defense-marker gene, *PATHOGENESIS-RELATED 1* (*PR1*) in *Arabidopsis* leaves (Supplementary Fig. 1). Proteinase-K treatment abolished RsE immunogenic activity, suggesting that the elicitor corresponds to one or more peptides or proteins (Supplementary Fig. 2a). In gel-filtration experiments, the molecular mass of RsE elicitor activity was estimated to be <10 kDa (Supplementary Fig. 2b).

To identify the RsE receptor, we screened a collection of 106 natural strains, or accessions, of *Arabidopsis* for RsE-induced ethylene production (Supplementary Fig. 3). Three accessions (Dog-4, ICE21, and ICE73) with reproducibly strongly reduced ethylene production relative to that of reference accession Col-0 were deemed RsE-insensitive and selected for in-depth analysis (Fig. 1a and Supplementary Fig. 3). These accessions remained sensitive to SCLEROTINIA CULTURE FILTRATE ELICITOR 1 (SCFE1), an unrelated fungal elicitor recognized by RLP30[30] (Fig. 1a), suggesting that insensitivity to RsE was not due to a general defect in defense activation. Approximately 60% of the tested accessions produced more ethylene than Col-0 in response to RsE, including hypersensitive accession ICE153 (Fig. 1a and Supplementary Fig. 3).

Insensitive accessions were crossed reciprocally (Fig. 1b and Supplementary Fig. 4). F1 progeny from all crosses remained RsE-insensitive, suggesting that changes at the same locus render these accessions insensitive to RsE. Crossing insensitive accessions with

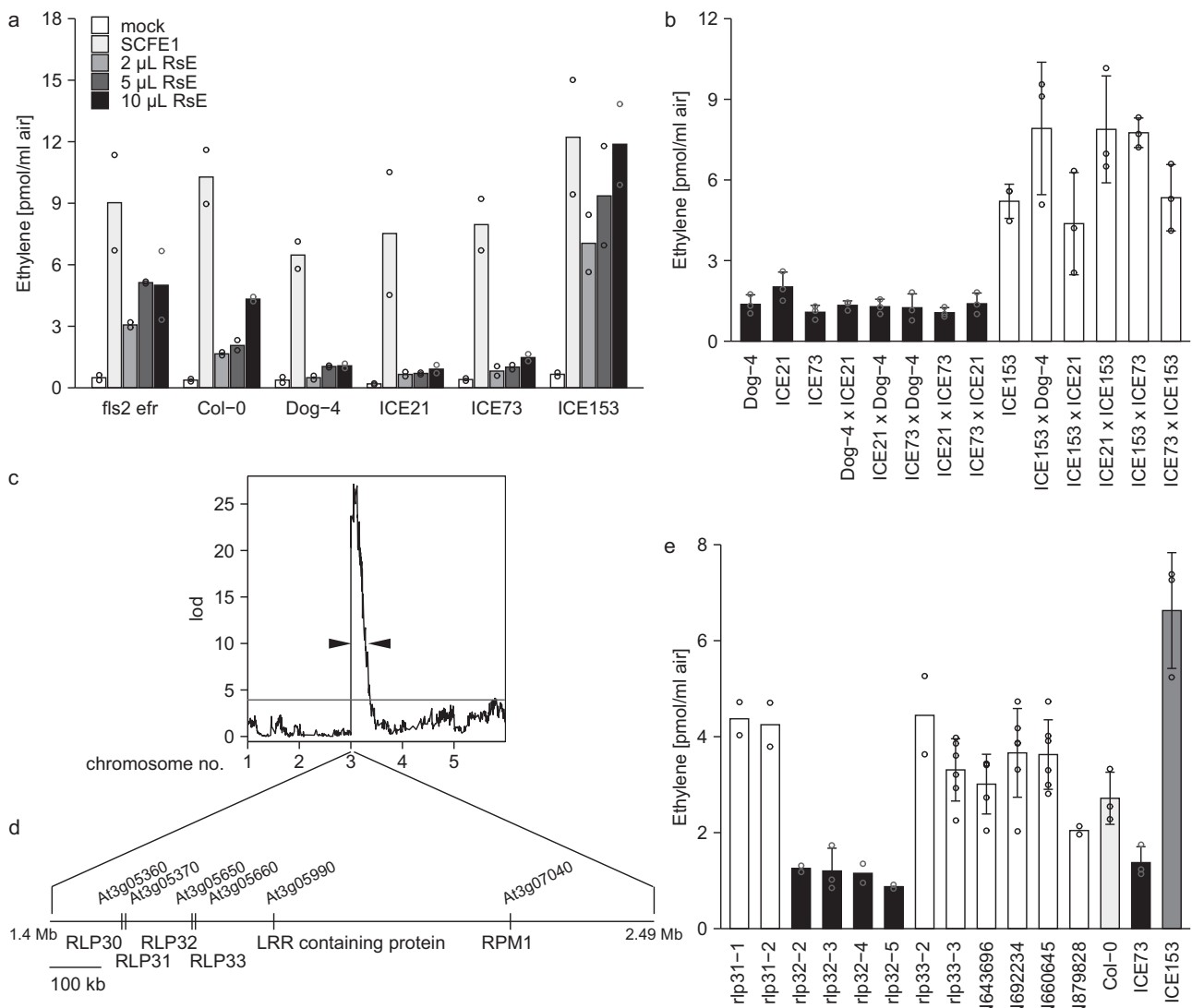

**Fig. 1 RAD-seq-associated QTL analysis-based identification of the RLP32 locus conferring sensitivity to *R. solanacearum* elicitor RsE. a** Ethylene production in *Arabidopsis* Col-0 wild-type, *fls2 efr* mutant, and RsE-insensitive (Dog-4, ICE21, and ICE73) and RsE-hypersensitive (ICE153) accessions. Treatments with water (mock) and SCFE1 served as controls. Data points indicate two replicates. **b** RsE-induced ethylene production in crosses of sensitive and insensitive accessions as described in Supplementary Fig. 4. Black bars indicate insensitive-accessions or a cross of two insensitive accessions, and white bars indicate sensitive accession ICE153 or its crosses with insensitive accessions. Shown are mean values of three replicates ± SD. **c** QTL mapping for RsE-induced ethylene response in $F_2$-mapping populations of an ICE153 x ICE73 cross. LOD scores from a full-genome scan across five chromosomes of *Arabidopsis* using a binary-trait model for RsE-elicited ethylene scores were plotted. The horizontal line indicates the significance threshold (an expectation-maximization (EM) algorithm was applied with a permutation of 1000 repeats and type-I error-rate alpha = 0.05). Arrow heads indicate a region with LOD value above 10. **d** Genomic arrangement of selected candidates within a 1.1 Mb region on chromosome 3 identified by QTL analysis as putative loci conferring RsE sensitivity. **e** RsE-induced ethylene production in *Arabidopsis* plants carrying T-DNA insertions in genes indicated in (**d**). Ethylene production in Col-0 wild type and accessions ICE73 and ICE153 served as controls. Shown are mean values ± SD (*n* = 3 for *rlp32-3*, Col-0, ICE73, and ICE153; *n* = 5 for *N643696*; *n* = 6 for *rlp33-3*, *N692234*, and *N660645*). *n* = 2 biologically independent samples for *rlp31-1*, *rlp31-2*, *rlp32-2*, *rlp32-4*, *rlp32-5*, and *N879828*.

hypersensitive ICE153 produced only RsE-sensitive $F_1$ plants (Fig. 1b). $F_2$ populations of an ICE153 x ICE73 cross exhibited a segregation ratio of 1:3 (92 insensitive plants vs. 303 sensitive plants), suggesting that RsE sensitivity segregates as a single, recessive Mendelian trait. Thus, natural variation in *Arabidopsis* RsE sensitivity could be employed to identify the corresponding PRR.

**Genotyping-by-sequencing-based identification of RLP32.** Restriction-site-associated DNA sequencing (RAD-seq) was conducted to identify the genomic region conferring RsE sensitivity in *Arabidopsis*. DNA samples of 84 RsE-insensitive and 108

RsE-sensitive $F_2$ plants of the ICE153 x ICE73 cross were digested with *Pst*I/*Mse*I, and the DNA fragments were used to generate a tagged library, which was single-end sequenced (Illumina HiSeq2000, 36,4-fold average coverage) to yield 16,973 potential markers. In total, 901 markers were selected for further quantitative trait locus (QTL) mapping. Whether using quantitative or binary traits (RsE-sensitive 0, RsE-insensitive 1), QTL analyses identified one major peak on chromosome 3 associated with RsE sensitivity, consistent with Mendelian segregation (Fig. 1c and Supplementary Fig. 5a). Confidence-interval analysis (LOD score 10) using a binary-trait model defined a QTL interval between markers Chr3:20373 and Chr3:6582498 (Supplementary Fig. 5b). Recombination-breakpoint analyses in $F_2$ populations derived

from the ICE153 × ICE73 cross were used to narrow down the QTL to a 1.1 Mb region flanked by markers Chr3:1399533 and Chr3:2485009 (Supplementary Fig. 6). This region contained 339 open-reading frames, six of which encoded LRR-type proteins as prime candidates for PRRs (RLP30, At3g05360; RLP31, At3g05370; RLP32, At3g05650; RLP33, At3g05660; LRR-containing protein At3g05990; disease-related R protein/RPM1, At3g07040) (Fig. 1d). We focused on the analysis of LRR proteins first, as proteinaceous immunogenic patterns, such as RsE (Supplementary Fig. 2a), are predominantly recognized by LRR ectodomain receptors. RLP30 was excluded from further analysis because RsE-insensitive accessions recognized SCFE1 (Fig. 1a) and because SCFE1-insensitive accession Bak-2 was responsive to RsE (Supplementary Fig. 3). Knockdown or knockout alleles of the remaining genes were tested for RsE-inducible ethylene production. Four independent T-DNA/transposon alleles of *RLP32* in the reference accession Col-0 (insertions validated by flanking-fragment sequencing; Supplementary Fig. 7), proved insensitive to RsE, suggesting that RLP32 confers RsE sensitivity (Fig. 1e). All other tested mutants responded to RsE (Fig. 1e).

Col-0 RLP32 is composed of an ectodomain comprising 23 LRR units with an island domain separating LRRs 19 and 20, a juxta-membrane domain, a transmembrane domain, and a 28-amino-acid tail (Supplementary Fig. 8). Inspection of RLP32 protein sequences from 93 accessions including those from 7 RsE-insensitive and 19 RsE-hypersensitive accessions (relative to RLP32 activity observed in Col-0) (Supplementary Fig. 3), revealed rather diverse polymorphisms among RLP32 orthologs (Supplementary Fig. 9), thus making predictions about mutations causal for loss of RsE sensitivity difficult. For example, RLP32 sequences from RsE-insensitive accession ICE21 and RsE-sensitive ICE71 differed in only three amino acids (ICE71 residues S204, V764, and S809). Likewise, RLP32 (ICE21) and RsE-sensitive RLP32 (Col-0) sequences differ by only four residues (Q116, D613, E764, and G837). However, polymorphisms observed in RLP32 (ICE21) are not conserved among RsE-insensitive accessions, and polymorphisms in RLP32 from Col-0 and ICE71 are not conserved in RsE-sensitive accessions (Supplementary Fig. 9). Because of these findings and because *RLP32* alleles from RsE-insensitive accessions are only distantly related (Supplementary Fig. 10), we conclude that loss of canonical RLP32 activity is determined by several unrelated sequence polymorphisms and may have occurred repeatedly during evolution.

**RLP32 recognizes proteobacterial translation-initiation factor 1 (IF1).** To assess whether RsE activity is found in bacteria other than *R. solanacearum*, protein extracts from *Escherichia coli*, which is not a plant pathogen, were fractionated using the protocol for RsE preparation. As shown in Supplementary Fig. 11a, *fls2 efr* mutant plants, but not *rlp32* mutants produced ethylene upon *E. coli* elicitor treatment, suggesting that RsE-like activity is not restricted to *R. solanacearum*.

Partially purified *E. coli* elicitor was subjected to liquid chromatography–mass spectrometry (LC–MS/MS)-based protein identification (Supplementary Fig. 11b). Of a total of 290 proteins identified, 20 proteins were chosen for further analysis according to the following criteria: (i) a high number of peptides with different masses found in all four elicitor-active fractions, (ii) a predicted molecular mass of RsE of approximately 10 kDa (Supplementary Fig. 2b), and (iii) a basic isoelectric point as deduced from the migration of RsE elicitor activity in ion-exchange chromatography experiments. Of the candidate proteins tested, only protein translation-initiation factor 1 (IF1) induced RLP32-dependent ethylene production (Fig. 2a and

Supplementary Fig. 11c). IF1 also triggered RLP32-dependent ROS production, deposition of callose, phosphorylation of MAPKs, as well as accumulation of *FRK1*, *PAD3*, and *PDF1.2* defense-marker gene transcripts (Supplementary Fig. 12).

IF1 is a 72-amino-acid single-domain protein composed of a five-stranded β-barrel and a short α-helical loop connecting strands 3 and 4 (Supplementary Fig. 13a, b). IF1 belongs to the oligonucleotide-binding-fold protein family and shares close structural homology to another member of this family, bacterial cold-shock protein CspA[31], a trigger of immunity in tobacco and tomato[32]. Recombinant *E. coli* IF1 triggers ethylene production at low nanomolar concentrations ($EC_{50} = 5.8$ nM) (Fig. 2b). Likewise, *E. coli* IF1 produced by in vitro transcription and translation or by chemical synthesis exhibited RLP32-dependent elicitor activity (Fig. 2c). This finding indicates that IF1, and not a copurifying contaminant, triggers RLP32-mediated plant defense.

IF1 amino-acid sequences are highly conserved among proteobacteria (Supplementary Fig. 13a). IF1 proteins from plant pathogenic *Pseudomonas syringae*, *Agrobacterium tumefaciens*, and *R. solanacearum* share with *E. coli* IF1 85%, 60%, or 75% sequence identity, respectively. Likewise, iterative threading-assembly refinement (I-TASSER)-based 3-dimensional structure prediction (https://zhanggroup.org/I-TASSER/) revealed strong secondary- and tertiary-structure conservation of plant patho-genic bacteria-derived IF1 molecules when compared with *E. coli* IF1 (Supplementary Fig. 13b, c). IF1 preparations from these bacterial species exhibited elicitor activities similar to that of *E. coli* IF1 (Fig. 2d and Supplementary Fig. 14a). Our findings suggest that IF1 is a widespread bacterial pattern and that *R. solanacearum* IF1 accounts for the elicitor activity in RsE preparations (Supplementary Fig. 14a). Synthetic IF1 based on genome sequences from *Lysobacter* or *Rhizobacter* strains that are associated with the *Arabidopsis* root microbiome[33] had substantially less elicitor activity than *E. coli* IF1, and *Arabidopsis* chloroplast-encoded IF1 neither exhibited elicitor activity in *Arabidopsis* itself nor in *N. benthamiana* plants transiently expressing RLP32 (Supplementary Fig. 14b–f), consistent with the absence of autoimmunity triggered by endogenous IF1.

Stable expression of *pRLP32::RLP32* constructs in *Arabidopsis rlp32* mutants and in the RsE-insensitive accession ICE73 conferred sensitivity to both RsE and IF1 (Fig. 3a, b). Likewise, overexpression of *p35S::RLP32-GFP* in RsE- and IF1-insensitive *N. benthamiana* conferred sensitivity to IF1 (Fig. 3c). Collectively, these data confirm that the active component of RsE is IF1 and that RLP32 mediates IF1-inducible defenses without the need for additional species-specific factors.

**Tertiary-structure properties are required for IF1-mediated plant-defense activation.** Immunogenic activities of large proteinaceous patterns are typically represented by short, conserved peptide fragments. We have tested chemically synthesized nested peptides spanning the entire IF1 protein sequence, as well as recombinantly expressed IF1 fragments carrying N-terminal or C-terminal deletions (Fig. 4a, b). Peptide fragments were designed to preserve IF1 secondary structural motifs (Supplementary Fig. 15). With the exception of a near full-length IF1 variant (I7–R72) with an N-terminal deletion of a short unstructured segment of six amino-acid residues (Supplementary Fig. 15), all IF1 peptide fragments failed to trigger RLP32-dependent ethylene production (Fig. 4a, b). These findings suggest that tertiary structure rather than primary or secondary-structure motifs determine IF1 elicitor activity.

Proteobacterial IF1 and cold-shock protein CspA share a highly conserved five-stranded β-barrel fold[31]. One notable difference is that the α-helical motif connecting IF1 β-strands 3 and 4 is absent in

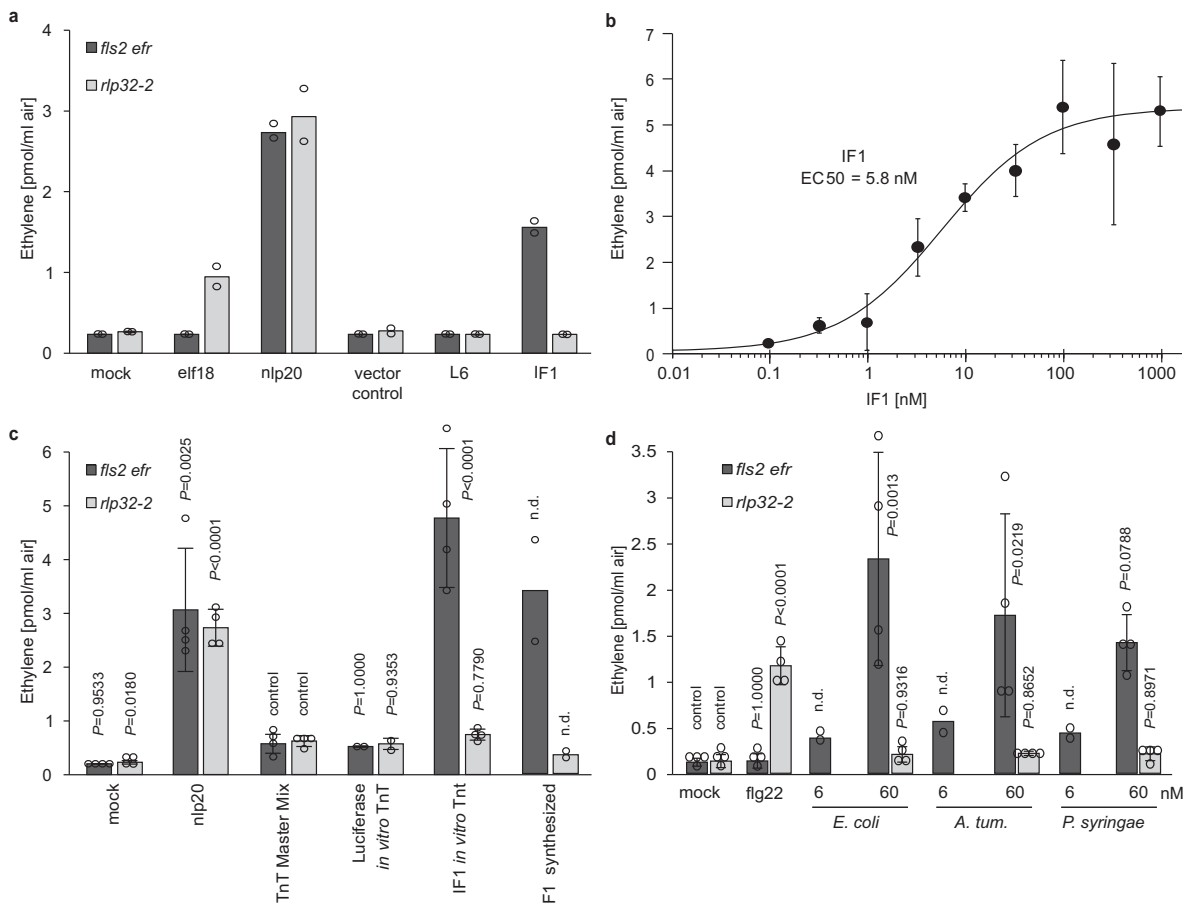

**Fig. 2 IF1 elicitor activity. a** Ethylene accumulation was determined in *Arabidopsis fls2 efr* or *rlp32* mutants treated with recombinant IF1 and ribosomal protein L6 (see Supplementary Fig. 11c), or with protein preparations from *E. coli* transformed with empty vector. Treatment with water (mock), elf18, or nlp20 served as controls. Data points indicate two biologically independent replicates. **b** Determination of EC$_{50}$ values using increasing concentrations of recombinant IF1 (produced in yeast). EC$_{50}$ values and curve fit were calculated using 4 P Rodbard Model comparison. **c, d** Ethylene accumulation in *Arabidopsis fls2 efr* or *rlp32* mutants treated with in vitro-translated (TnT) IF1, chemically synthesized IF1 (**c**), or recombinant IF1 from the bacterial species indicated. *A. tum, Agrobacterium tumefaciens*. (**d**). Bars represent means ± SD of three (**b**) or four (**c, d**, each pooled from two experiments) replicates. (n.d. not determined, two-sided Dunnett's test with mock (**d**) or TnT MasterMix (**c**) as control, respectively). Experiments were performed at least twice with similar results.

CspA. As *E. coli* CspA is not recognized by *Arabidopsis* (Supplementary Fig. 16), we hypothesized that this short helix is important for IF1 elicitor activity. Simultaneous replacement of positively charged residues K39, R41, and K42 by leucine residues or introduction of a proline residue, an amino acid known to distort helical structures because of forming a kink in the peptide backbone, did not reduce IF1 elicitor activity (Fig. 4c). Introduction of the IF1 helical motif into the 5-stranded β-barrel of *E. coli* CspA did not confer RLP32-dependent elicitor activity to the chimeric protein when tested in *Arabidopsis* (Fig. 4c). Hence, the IF1 helical motif is likely not important for IF1 elicitor activity. Heat treatment strongly reduced IF1 elicitor activity (Fig. 4d), suggesting that heat-induced alterations within the IF1 tertiary fold adversely affected its ability to trigger plant defense. Altogether, our findings suggest that tertiary-structure features rather than primary or secondary-structure motifs determine IF1 elicitor activity.

**IF1 and elf18 do not act additively**. IF1 and EF-Tu are distinct components of the proteobacterial protein-translation machinery that are perceived by *Arabidopsis* PRRs RLP32 (this study) and EFR[19], respectively. We tested whether simultaneous action of both patterns, a scenario likely occurring during attempted bacterial ingress, has additive or synergistic effects on plant-defense

activation. IF1 and EF-Tu fragment elf18 were applied either alone or in combination at concentrations below or corresponding to the EC$_{50}$ values of the respective patterns (Fig. 2b)[19]. However, neither additive nor synergistic effects on pattern-induced ethylene production were observed (Fig. 4e).

**RLP32 is required for IF1-induced plant immunity**. Pattern treatment primes plant immunity to subsequent infection by host-adapted, virulent plant pathogens. To test whether this also applies to IF1, we pretreated Col-0 plants with IF1 24 h before inoculation with virulent *Pseudomonas syringae* pv. *tomato* (*Pst*) strain DC3000 (Fig. 5a). We found that treatment with IF1, like the positive control nlp20, reduced *Pst*DC3000 growth 3 days post inoculation compared with mock-treated plants. This priming effect was abolished in two independent *rlp32* mutant lines (Fig. 5b, c), suggesting that IF1 recognition by RLP32 contributes to plant immune activation and reduced microbial proliferation on infected plants. No differences in bacterial titers were observed in nonprimed Col-0 and *rlp32* mutants when spray-inoculated with *Pst*DC3000 or a hypovirulent *Pst*DC3000COR⁻ strain[34] (Supplementary Fig. 17a, b).

*Nicotiana benthamiana* plants are insensitive to IF1 treatment. However, stable transformation with *p35S::RLP32-GFP* conferred

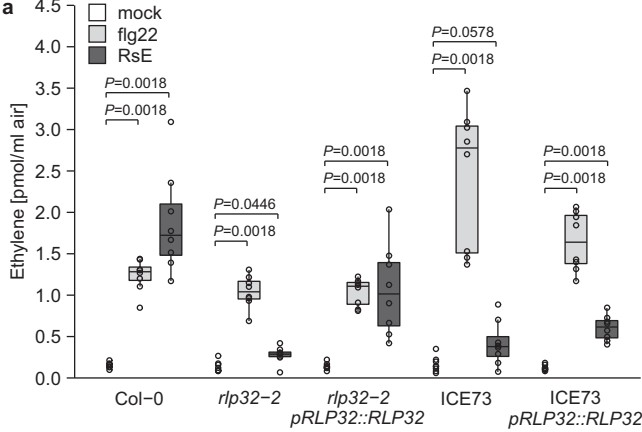

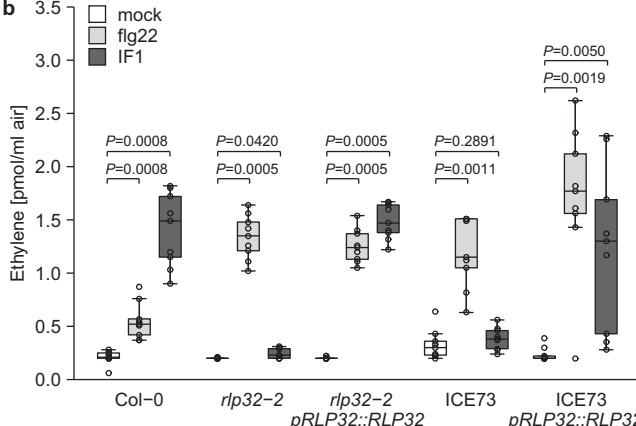

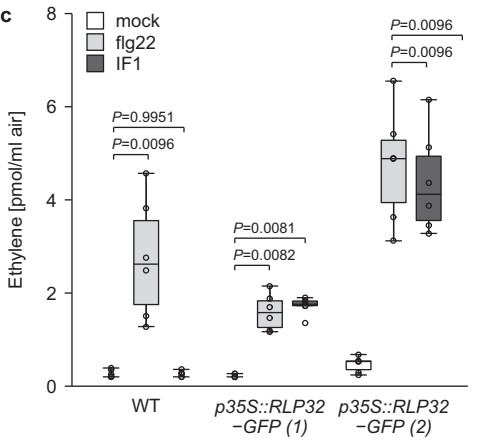

**Fig. 3 RLP32 confers sensitivity to RsE and IF1. a–c** Ethylene accumulation after treatment with flg22, IF1 (expressed with 6xHis in *E. coli*), RsE, or water (mock) in *Arabidopsis rlp32* mutants, RsE-insensitive accession ICE73, *Arabidopsis* plants stably expressing p*RLP32::RLP32* (**a**, **b**), and in two independent p*35S::RLP32-GFP* transgenic *N. benthamiana* lines (**c**). The center lines of the box plots indicate the median, the bounds of the box show the 25th and the 75th percentiles, and the whiskers indicate 1.5 × IQR. (**a**, *n* = 8; **b**, *n* = 9; **c**, *n* = 6, pooled from each three to four experimental repetitions; two-sided Steel test with mock treatment as control).

IF1 sensitivity to independent transgenic lines (Fig. 3c). When inoculated with bacterial strains *Pst*DC3000Δ*hrcC*[35] or *P. syringae* pv. *tabaci*[36], transgenic plants supported less bacterial growth compared with bacterial growth in untransformed control plants (Fig. 5d, e). In contrast, RLP32-expressing *N. benthamiana* plants did not restrict growth of *Pst*DC3000Δ*hopQ1–1*[37] (Supplementary Fig. 17c).

**RLP32 binds IF1 and forms a PRR complex with co-receptors SOBIR1 and BAK1.** Biologically active, biotinylated IF1 (bio-IF1) was used to analyze binding to RLP32 *in planta* (Fig. 6a and Supplementary Fig. 18). Leaves of p*35S::RLP32-GFP*-expressing *N. benthamiana* plants were treated with bio-IF1 before infiltration of the homobifunctional chemical cross-linker, EGS (ethylene glycol bis-succinimidyl succinate). RLP32-GFP was subsequently precipitated with GFP-trap beads and analyzed for ligand binding using a streptavidin–alkaline phosphatase conjugate. Control experiments were conducted using biotinylated nlp24 (nlp24-bio) cross-linked to transiently expressed RLP23-GFP[17]. In both cases, ligand binding to the respective receptors was observed at concentrations similar to those required for pattern-induced ethylene production (Fig. 6a). Loss of bio-IF1 binding to RLP32 in the presence of a 1000-fold molar excess of native IF1 demonstrated ligand specificity of this binding event (Fig. 6a). Taken together, our findings suggest that RLP32 is a sensor for proteobacterial IF1.

LRR-RP-type PRRs constitutively interact with SOBIR1 and recruit BAK1 into a receptor–ligand complex in a ligand-dependent manner. We conducted co-immunoprecipitation assays in transiently transformed *N. benthamiana* plants to demonstrate elicitor-induced formation of RLP32–SOBIR1–BAK1 complexes. As shown in Fig. 6b, RLP32 SOBIR1 complexes were formed independently of IF1 treatment. In contrast, RLP32 SOBIR1 BAK1 complexes were formed only in elicitor-treated plants (Fig. 6b). *Arabidopsis sobir1* and *bak1–5/bkk1–1* mutants proved insensitive to IF1-induced ethylene production, thus confirming a role of SOBIR1 and BAK1 in RLP32-mediated plant defense (Fig. 6c).

**Accession-specific differences in IF1 responsiveness are not linked to RLP32 protein-sequence polymorphisms.** Substantial differences in RsE sensitivities were observed among the 106 *Arabidopsis* accessions tested in our initial screen (Supplementary Fig. 3). Some accessions exhibited lower levels of RsE-induced ethylene production relative to that in accession Col-0, whereas the majority of the accessions tested showed higher ethylene production. RsE-hypersensitive accessions are also more sensitive to IF1 than accession Col-0 (Supplementary Fig. 19a). However, when treated with flg22 or nlp20 patterns, RsE/IF1-hypersensitive accessions also showed increased levels of ethylene production relative to those observed in elicitor-treated Col-0 (Supplementary Fig. 19a). These findings suggest that RsE/IF1 hypersensitivity is not due to accession-specific RLP32-sequence information. In agreement with this notion, transient expression of *RLP32* alleles from Col-0- and IF1-hypersensitive accessions ICE93 and ICE71 (Supplementary Fig. 3) in *N. benthamiana* resulted in similar levels of IF1-induced ethylene production (Supplementary Fig. 19b).

**IF1 sensitivity is restricted to selected Brassicaceae.** To assess the distribution of IF1-recognition systems among plants, we tested IF1-inducible ethylene production in close relatives of *Arabidopsis* (Fig. 6d). *Capsella rubella* and *Arabis alpina* lacked IF1 sensitivity, but *Brassica rapa* and *B. oleracea* responded to IF1 treatment. In contrast, a breeding variant of *B. oleracea*, *B. oleracea* var. *botrytis*, proved insensitive to IF1 treatment. Likewise, members of the *Solanaceae* (*Nicotiana benthamiana*, *Nicotiana tabacum*, and *Solanum pennellii*) did not recognize IF1 (Fig. 6d). IF1 sensitivity appears to be rare, or even absent in these species.

**Discussion**

Here we report biochemical purification and mass spectrometry-based identification of bacterial translation-initiation factor 1 (IF1), we characterize its immunogenic activity in *Arabidopsis*

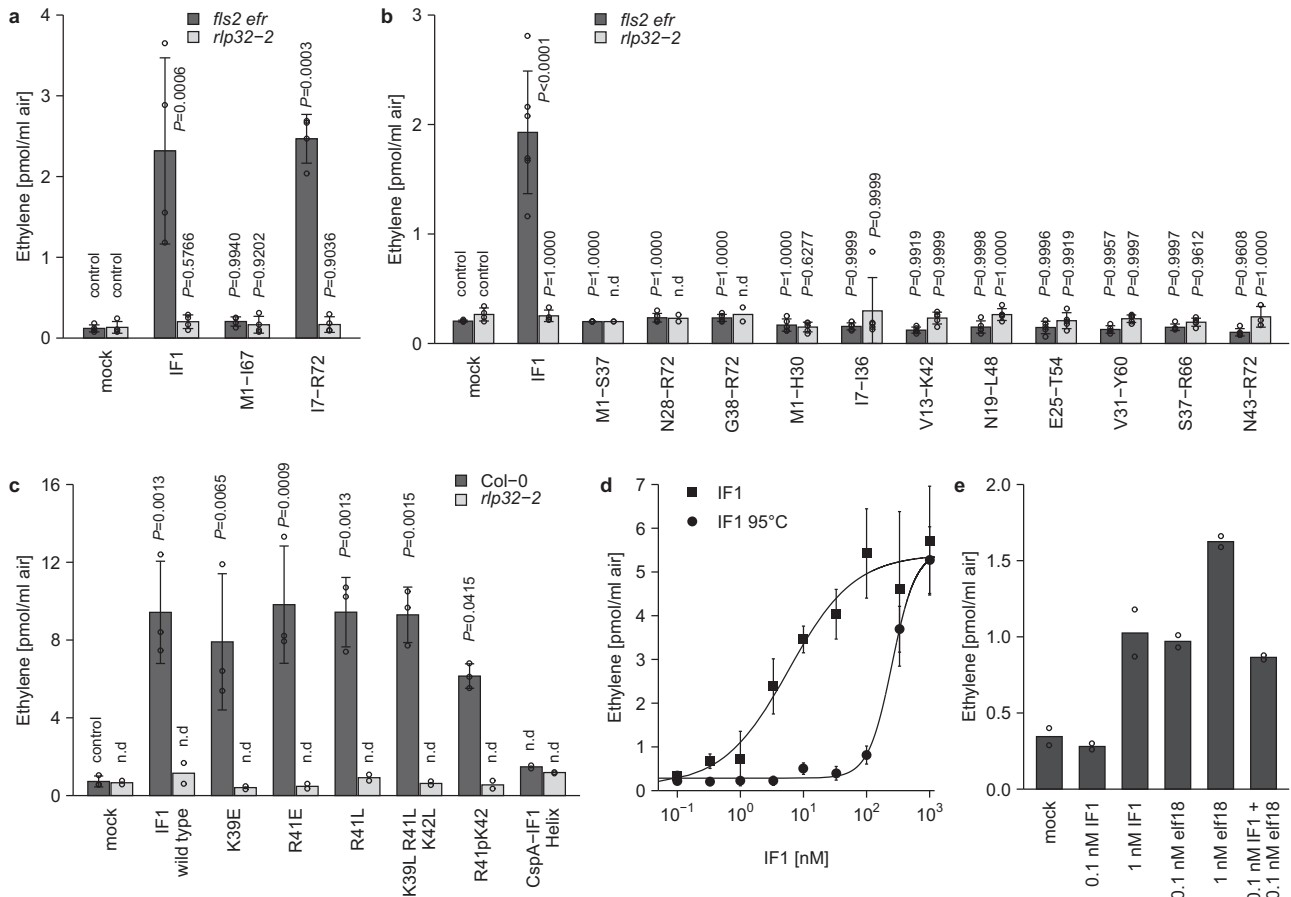

**Fig. 4 IF1 tertiary-fold features are required for its elicitor activity. a** Ethylene accumulation in *Arabidopsis fls2 efr* or *rlp32-2* mutants treated with 60 nM IF1 or with 40 nM IF1_I7–R72 and IF1_M1–I67 (produced in *E. coli*). **b** Ethylene accumulation after treatment with IF1 or indicated synthetic IF1 variants. **c** Ethylene accumulation after treatment with IF1 produced in yeast, indicated IF1 point mutants or a CspA–IF1 helix chimeric protein. **d** Determination of EC$_{50}$ values using increasing concentrations of IF1 produced in yeast before and after heat treatment. Note that the IF1 data set has been taken from Fig. 2b. **e** Ethylene accumulation in *Arabidopsis* Col-0 wild-type plants treated with IF1 and elf18 alone or in combinations indicated. Bars represent means ± SD (**a**, n = 4; **b**, data points indicate the number of biologically independent replicates used. Exact *n*-values for each sample are provided in the Source Data file; **c**, n = 2 for *rlp32-2*, n = 3 for Col-0; **d**, n = 3, **e**, n = 2; **a**, **b**, pooled data from each of the two experiments, two-sided Dunnett's test with mock treatment as control).

and related Brassicaceae species, and we identify RLP32 as an IF1 receptor. IF1, an 8.2 kDa polypeptide, is likely one component within a low-molecular-weight protein fraction from *R. solanacearum* that was previously shown to trigger immunity in *Arabidopsis* in an FLS2-independent manner[29]. IF1 molecules from taxonomically unrelated proteobacteria are active inducers of *Arabidopsis* defense (Fig. 2d), which is in agreement with their high primary- and tertiary-structure conservation (Supplementary Fig. 13). Generally, IF1 conforms to the definition of classical PAMPs as immunogenic molecules that are (i) ubiquitous to whole classes of microbes, (ii) structurally conserved across microbial species or genus boundaries, and (iii) not found in potential host organisms.

Bacterial IF1 and translation-elongation factor EF-Tu constitute not only different parts of the same molecular machineries, but share similar immunogenic activities in *Arabidopsis* (Fig. 4e). During attempted plant infection, both patterns are likely to trigger immune defenses simultaneously. However, concomitant application of both patterns at nonsaturating concentrations did not result in additive or synergistic increases in plant-defense output (Fig. 4e). Hence, plants may integrate various external stimuli to mount an appropriate output response, suggesting that signal input may not necessarily be directly linked to plant-defense outputs.

Immunogenic activities of virtually all known microbe-derived plant-defense elicitors can be ascribed to small epitopes within these molecules[1]. IF1 appears to be a remarkable exception to this rule as our collective experimental efforts suggest that the IF1 tertiary fold is required for its elicitor activity. Testing a library of nested peptides spanning IF1 or of peptides covering individual IF1 secondary-structure motifs failed to reveal a small peptide elicitor (Fig. 4a, b). Likewise, peptide mixtures and larger terminal-deletion mutants affecting IF1 secondary-structure motifs lacked elicitor activity (Fig. 4a, b). IF1 shares with elicitor-inactive bacterial cold-shock protein CspA a highly conserved 5-strand β-barrel fold[31], and carries an additional short α-helical motif between β-strands 3 and 4. However, introduction of single or higher-order mutations into this helical motif did not affect IF1 elicitor activity, nor did engineering of the IF1 helix into CspA (CspA–IF1 helix) result in an active elicitor (Fig. 4c). Together with the observed heat instability of IF1 activity (Fig. 4d), our data suggest that tertiary-structure features rather than primary-sequence motifs determine IF1 elicitor activity. While apparently uncommon for plant immunogenic patterns, structural fold requirements for the activation of pattern-induced immunity have been reported from metazoans. For example, activation of human TOLL-LIKE RECEPTOR 5 (TLR5) is brought about by recognition of large internal helical structures within intact flagellin[38].

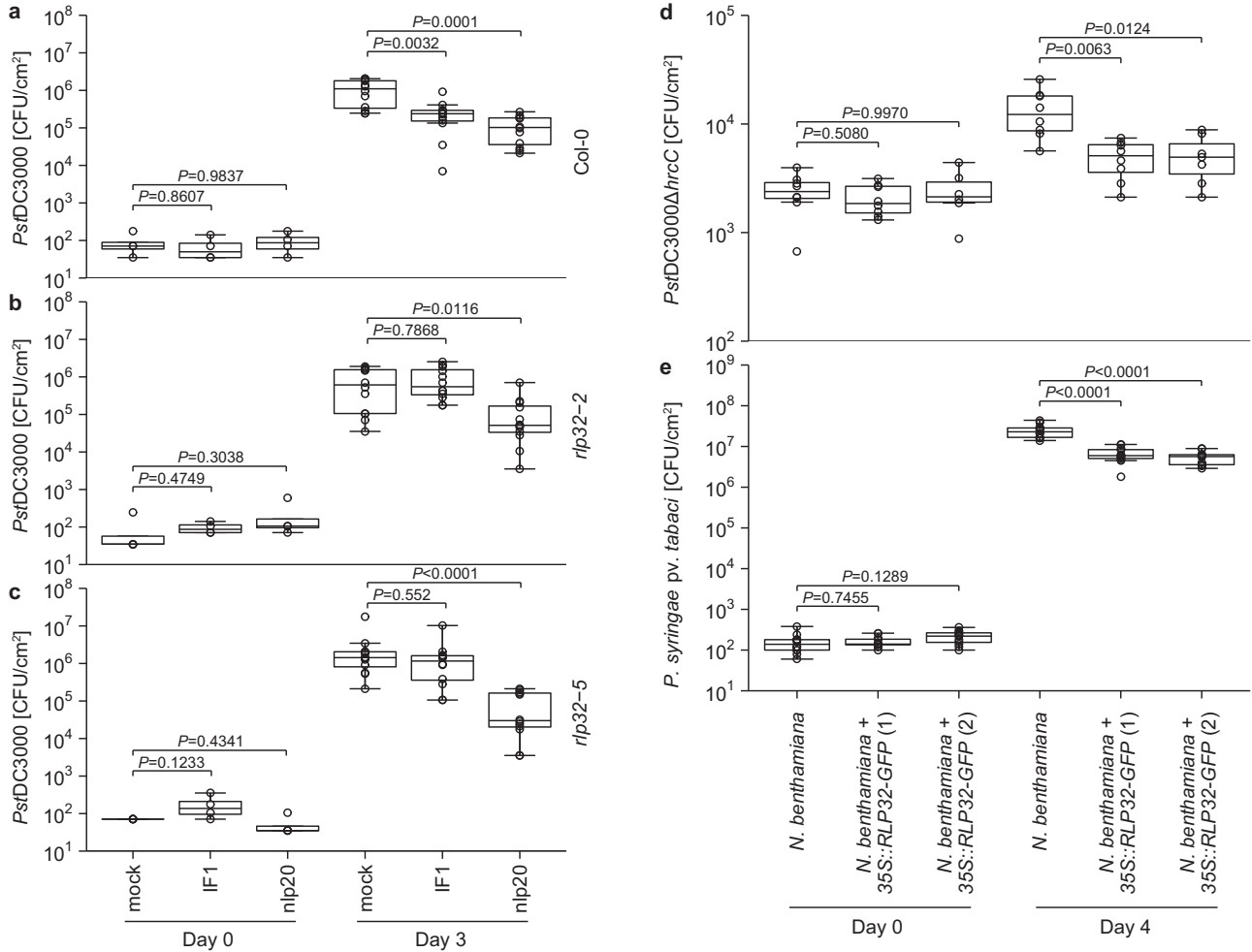

**Fig. 5 RLP32 mediates bacterial resistance. a–c** Growth of *Pseudomonas syringae* pv. *tomato* DC3000 (*Pst*) in *A. thaliana* Col-0 (**a**) and *rlp32* plants (**b**, **c**) after priming with IF1 or nlp20 24 h prior to bacterial inoculation (syringe infiltration). Water infiltration served as a control (mock). Bacterial growth was determined 0 and 3 days post inoculation. **d, e** Growth of *Pst*DC3000Δ*hrcC* (**d**) or *P. syringae* pv. *tabaci* (**e**) in wild-type or two *RLP32*-transgenic *N. benthamiana* lines at days 0 and 4 after inoculation. For the box plots, the center line indicates the median, the bounds of the box show the 25th and the 75th percentiles, and the whiskers indicate 1.5 × IQR (**a–c**, n = 4 from 2 plants for day 0 and n = 12 from 6 plants for day 3; **d**, n = 6–8 from 4 plants, exact n-values are provided in the Source Data file; **e**, n = 12 from 6 plants; two-sided Steel test with mock treatment as control; no significant differences were observed for day 0). Experiments were performed three times, with similar results.

We have exploited natural variation between *Arabidopsis* accessions to identify the IF1 receptor. A continuous spectrum of phenotypic variation of IF1 sensitivity limited the power of GWAS (genome-wide association studies) studies (Supplementary Fig. 3). The use of heterogeneous RsE fractions, which could potentially contain multiple elicitors, further increased the risk of ambiguous phenotype determination in segregating populations. These hindrances would also prove challenging to classical PCR polymorphism marker-based mapping approaches that are low-throughput and thus require repetitive phenotyping. To avoid laborious marker testing and phenotyping errors inherent to map-based cloning, we employed a genotyping-by-sequencing (GBS) approach to identify and score approximately 900 markers in a nonreference biparental population. QTL analysis of the $F_2$-segregating population derived of representative extreme morphs enabled the identification of a sizable genomic interval in chromosome 3 encoding RsE sensitivity (Fig. 1c and Supplementary Figs. 5 and 6). Subsequent RAD-seq-based identification of recombination-break boundaries from the very same sample set

narrowed down the RsE-encoding region to a 1.1 Mb fragment encoding a small number of receptor candidates (Fig. 1d). This confirms that RAD-seq-based QTL mapping can be superior to classical mapping approaches as linking particular phenotypes to defined genomic loci is supported by strong statistical analyses due to the use of large marker sets[39].

Reverse genetics analysis identified RLP32 as a locus conferring RsE/IF1 sensitivity. Subsequent genetic and biochemical assays established a role of RLP32 as the IF1 receptor. Evidence for this is based on the following findings: (1) *rlp32* mutants do not mount IF1-inducible defenses, (2) ectopic expression of RLP32 in IF1-insensitive *Arabidopsis* accessions and in *rlp32* mutants confers IF1 sensitivity, (3) production of RLP32 in IF1-insensitive *N. benthamiana* confers IF1 sensitivity, (4) RLP32 specifically binds IF1, (5) RLP32 forms with SOBIR1 and BAK1, a ternary immune-receptor complex similar to that known for other *Arabidopsis* LRR-RP-type PRRs, and (6) wild-type plants, but not *rlp32* mutants restrict bacterial growth following pretreatment with IF1.

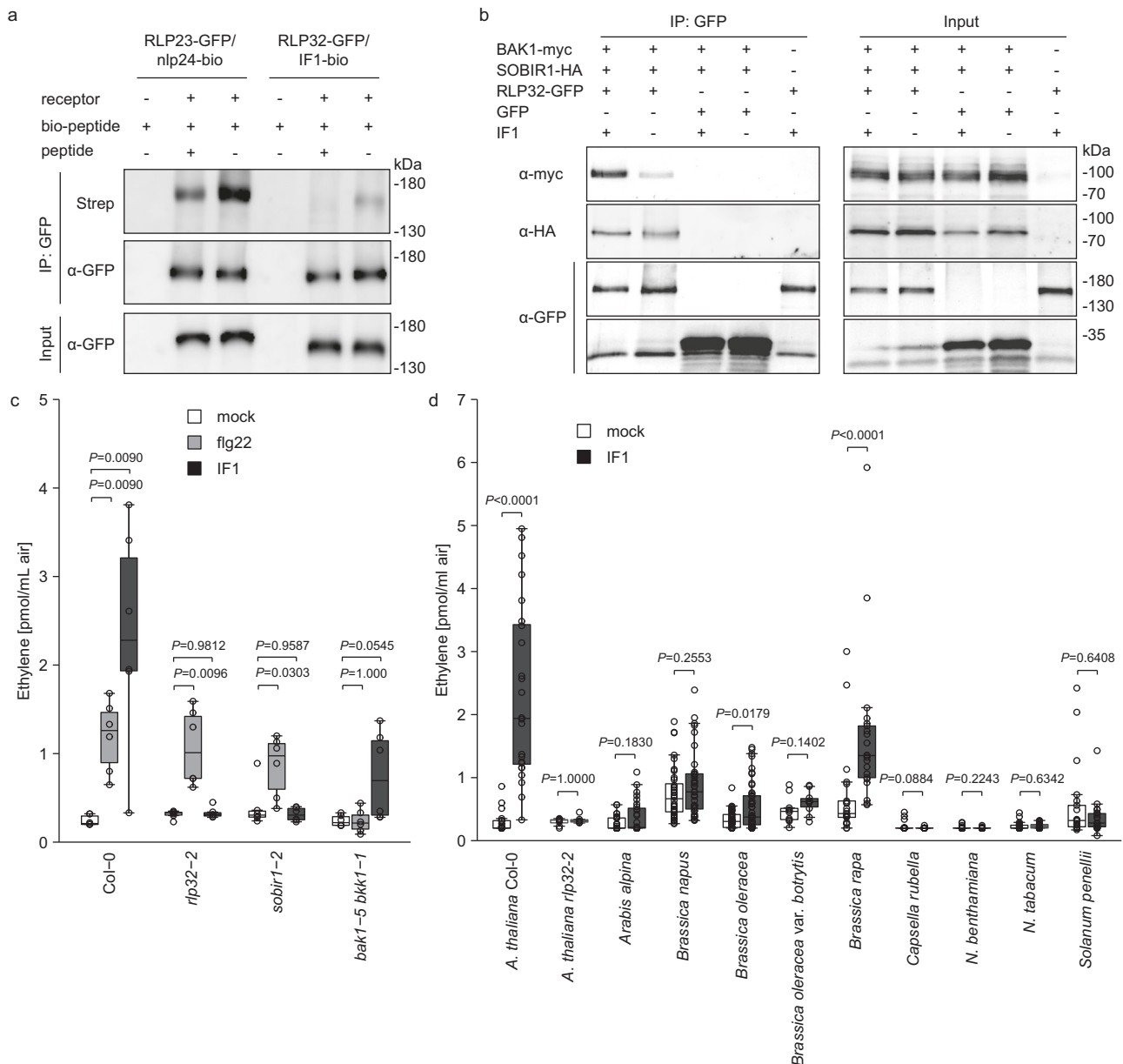

**Fig. 6 RLP32 binds to IF1 and forms complexes with SOBIR1 and BAK1. a** Protein-blot analysis of cross-linking assays using GFP-trap-purified (IP:GFP) proteins obtained from *RLP32-GFP* transgenic *N. benthamiana* plants (input) treated with 30 nM biotinylated IF1 (IF-bio) as ligand. A 1000-fold molar excess of unlabeled peptide (peptide) was used as competitor of ligand binding. Receptor-bound IF1-bio was visualized using a streptavidin–alkaline phosphatase conjugate (Strep). Transiently expressed RLP23-GFP treated with biotinylated nlp24 (nlp24-bio) and unlabeled nlp20 as competitor (peptide) served as controls. **b** RLP32-GFP, BAK1-myc, and SOBIR1-HA proteins were transiently expressed in *N. benthamiana* and treated with 1 μM IF1 (+) as indicated. Protein extracts (input) were subjected to co-immunoprecipitation using GFP-trap beads (IP: GFP), and bound proteins were analyzed by protein blotting using tag-specific antisera. **c** Ethylene accumulation in *Arabidopsis* Col-0 wild-type plants or indicated mutants after treatment with 500 nM IF1 or flg22 (*n* = 6, two-sided Steel's test with mock treatment as a control). **d** Ethylene accumulation in indicated plant species after treatment with IF1 (*n* = 6–48, pooled from four experiments, two-sided Mann–Whitney-U-test, exact *n*-values are provided in the Source data file). For the box plots, the center line indicates the median, the bounds of the box show the 25th and the 75th percentiles, and the whiskers indicate 1.5 × IQR. Experiments were performed twice (**a**, **c**, **d**) or three times (**b**), with similar results.

Distribution of LRR-RP-type PRRs is remarkably restricted to individual plant genera. Functional homologs of *Arabidopsis* RLP1, RLP23, and RLP42 are only found in this genus as well as in a few related Brassicaceae[17,40,41]. Likewise, a tomato receptor for fungal xylanase (EIX2), *N. benthamiana* sensors for CspA (CSPR, CORE), and a fungal hydrolase (RXEG1) exhibit genus-specific distribution[21,22,42,43]. RLP32 is no exception to this rule as IF1 sensitivity appears rare outside Brassicaceae (Fig. 6d) and shows significant within-genus variation (Supplementary Fig. 3).

In conclusion, a rather genus-specific distribution as well as significant accession-specific sequence polymorphisms among *Arabidopsis* PRRs suggest highly dynamic evolution of PTI sensors in this plant.

Identification of the ligand-receptor pair IF1/RLP32 illustrates that recognition of a single-pathogen species by a given host plant can be enormously complex. In addition to RLP32, *Arabidopsis* has evolved at least five additional receptor systems to sense the bacterial pathogen, *P. syringae*. These receptors comprise well-

studied FLS2 and EFR, as well as sensors for bacterial mc-3-OH-fatty acids (LORE), peptidoglycan (LYM1–LYM3–CERK1), and xanthine/uracil permease (XPS1)[16,18,19,23,24]. We can only speculate why a single plant species may have evolved such a number of redundant microbial sensing systems. It seems to be reasonable to assume that plant PRR complexity may not be brought about by co-evolution of these two organisms alone, but may result from multiple independent evolutionary processes that were driven by exposure to various microbial threats. Redundancy in *Arabidopsis* PRRs may explain accession-specific losses of recognition specificities that have been reported for most microbial sensor systems in this plant.

## Methods

**Plant materials and growth conditions**. All plants, except for *Solanaceae*, were grown on soil for 5–6 weeks under standard conditions (150 µmol cm$^{-2}$ s$^{-1}$ light for 8 h, 40–60% humidity, 22 °C). *Arabidopsis* accession Col-0 was the background for all mutants used in this study: *bak1–5 bkk1–1*[44], *fls2 efr*[45], *rlp32-2* (SM_3_33092, N119803), *rlp32-3* (SALK_137467C, N657024), *rlp32-4* (SM_3_33695, N120406), *rlp32-5* (SM3_15851, N106446), *rlp31-1*[46], *rlp31-2*[46], *rlp33-2*[46], *rlp33-3*[46], *sobir1–12*[47], SALK_143696 (N643696, *At3g05990*), SALK_203784C (N692234, *At3g05990*), SALK_146601C (N660645, *At3g07040*), and SAIL_918_H07 (N879828, *At3g07040*). Seeds of T-DNA or transposon-insertion lines were purchased from ABRC or NASC, respectively. Insertions of these alleles were confirmed by comparing flanking sequences according to the Col-0 reference genome. For *rlp32* mutants, insertions were additionally verified by flanking-fragment sequencing (Supplementary Fig. 7) using the Phire Plant Direct PCR Kit (Thermo Fisher Scientific) and primers listed in Supplementary Table 1. *Arabidopsis* natural germplasms used in this study are part of the 80 resequenced accessions (ABRC CS76427)[48] and the Nordborg collection[49]. *N. benthamiana*, *N. tabacum*, and *Solanum pennellii* plants were grown for 4–5 weeks on soil in the greenhouse (16 h light, 60–70% humidity, 22 °C). For stable transformation, *N. benthamiana* plants were grown for 7–8 weeks in sterile culture on MS medium containing 2% sucrose (13 h light, 23 °C). *Arabidopsis* and *N. benthamiana* plants used for bacterial-infection assays were grown under a translucent cover on soil under standard conditions (150 µmol cm$^{-2}$ s$^{-1}$ light for 8 h, 40–60% humidity, 22 °C).

**Elicitors used in this study**. Synthetic IF1 protein and all IF1 mutant peptides, as well as nlp20[50], nlp24-bio[17], flg22[51] and csp22[32], were purchased from Genscript Inc. (Piscataway, New Jersey, US) and were initially dissolved in 100% DMSO at a concentration of 10 mM. For working stock solutions, flg22 was diluted in 0.1% BSA, 0.1 M NaCl to 1 µM, full-length IF1 was diluted in sterile-filtered 10 mM MES, pH 5.7, or in 100% DMSO to 100 µM, IF1 M1–S37 was diluted in 3% ammonium water to 1 mM, IF1 G38–R72 and IF1 N28–R72 were diluted in water to 1 mM. All working dilutions were prepared in water prior to use. If not stated otherwise, synthetic elicitors used in this study were applied at 1 µM concentrations. SCFE1 was used at a concentration of 0.25 µg/ml[30]. As elicitor activities and protein contents varied among RsE-containing fractions, sample volumes ranging from 1 to 20 µl were used to detect RsE elicitor activities.

**RsE preparation and LC–MS/MS analysis**. RsE was purified from *Ralstonia solanacearum* GMI1000[52] cells cultivated in Kelman medium (10 g/l glucose, 10 g/l peptone, and 1 g/l casein hydrolysate, pH 6.5)[53] at 28 °C for 36–48 h at 200 rpm[39]. In total, 5 l cell cultures were boiled, cooled on ice, and centrifuged at 5000 *g* for 15 min. For protein precipitation, ammonium sulfate was added to the supernatant to 90% saturation, and precipitated proteins were collected by centrifugation at 10,000 *g* and then redissolved in 50 mM MES, pH 5.2. The crude extract was dialyzed overnight at 4 °C in 50 mM MES, pH 5.2 (ZelluTrans, Carl Roth, or Slide-A-Lyzer Dialysis Cassette, Thermo Fisher Scientific, each 3.5 kDa MWCO) before loading onto a cation exchange HiTrapSP FF column (Cytiva, Uppsala, Sweden) using an Äkta explorer system (GE Healthcare, loading speed 1 ml/min) with buffers A (50 mM MES, pH 5.2) and B (50 mM MES, 0.5 M KCl, pH 5.2). Bound proteins were eluted with 100% buffer B, and pooled fractions were dialyzed in 50 mM Tris/HCl, pH 8.5. Anion exchange chromatography on dialyzed extracts was performed using a HiTrapQ FF column (GE Healthcare) with buffer C (50 mM Tris/HCl, pH 8.5) and D (50 mM Tris/HCl, 0.5 M KCl, pH 8.5). The flow-through was dialyzed in 50 mM MES, pH 5.2, before loading onto a Source 15 S 4.6/100 PE cation-exchange column (GE Healthcare) for high-resolution protein separation at a loading speed of 1 ml/min, followed by gradient elution with buffer B. Eluted fractions were tested for ethylene-inducing activity in *Arabidopsis fls2 efr* mutant plants, and active fractions were pooled and stored at −20 °C.

For LC–MS/MS analysis, active fractions purified via cation-exchange chromatography were further purified by reverse-phase high-pressure liquid chromatography (HPLC) using a C8 column (ZORBAX 300SB, 5 µm, 4.6 × 150 mm, Agilent, Waldbronn, Germany) and a gradient of buffers E (H$_2$O, 0.1% TFA) and F (isopropanol, 0.1% TFA) at a flow rate of 1 ml/min. Fractions

containing ethylene-inducing activity were analyzed by LC–MS/MS as described[17]. Proteins were digested with trypsin and the resulting peptides desalted with C18 StageTips. LC–MS/MS analysis was conducted on an EasyLC nano-HPLC coupled to an LTQ Orbitrap XL mass spectrometer. Peptides were separated by a 127-min stepwise gradient (5–33–50–90% solvent B containing 80% acetonitrile, 0.1% formic acid) at a flow rate of 200 nl/min. Collision-induced dissociation (CID) was employed to sequentially fragment the ten most intense precursor ions in each scan cycle. In all measurements, sequenced precursor masses were excluded from further selection for 90 s. Target values for MS/MS fragmentation were 5000 charges and 106 charges for the MS scan. MaxQuant software version 1.5.2.8 with integrated Andromeda search engine was used for data analysis. Escherichia coli sequences deposited in Uniprot were used for data mapping. Trypsin was defined as a protease with a maximum of two missed cleavage sites. Methionine oxidation and N-terminal acetylation were specified as variable modifications. Cysteine carbamidomethylation was set as fixed modification. Initial maximum mass tolerance allowed was set to 4.5 ppm (for survey scans) and 0.5 Da (for CID-fragment ions). Peptide, protein, and modification-site identifications were reported at a false-discovery rate (FDR) of 0.01 as estimated by a target/decoy approach. The mass spectrometry proteomics data have been deposited to the ProteomeXchange Consortium via the PRIDE partner repository with the dataset identifier PXD031124.

**RAD-seq and QTL analysis**. Restriction site-associated DNA sequencing (RAD-seq) and quantitative trait locus mapping were conducted using the F$_2$ mapping population of an *Arabidopsis* ICE153 × ICE73 cross[39]. Individual DNA samples were extracted from 192 plants of the F$_2$ population. DNA samples were double-digested with restriction enzymes *Mse*I and *Pst*I-HF. Barcode adapters with the *Pst*I-overhang were generated by annealing two single-strand oligos (5′-cacgacgc tcttccgatctNNNNNtgca-3′ and 5′-NNNNNagatcggaagagcgtcgtg-3′) in two 96-well plates. The *Mse*I-adapter was generated by annealing two other single-strand oligos (5′-taagatcggaagagcggggactttaagc-3′ and 5′-gatcggtctcggcattcctgctgaaccgctcttcc-gatct-3′). The digested DNA samples were then barcoded through ligation with the barcode adapter and *Mse*I-adapter in two 96-well plates. 192 ligation products from two plates were pooled together and served as PCR template. The DNA library was amplified using primers 5′-aatgatacggcgaccaccgagatctacactcttccctacacgacgctcttcc gatct-3′ and 5′-caagcagaagacggcatacgagatcggtctcggcattcctgctgaa-3′. Amplified DNA products with 350–500 bp were sequenced on an Illumina HiSeq2000 Sequencing System. The sequencing data were imported into SHORE software package (https://1001genomes.org/software/shore.html) with a maximum of two mismatches of barcodes. The short sequence reads were mapped to the reference genomes using Burrows-Wheeler Aligner (BWA) software (http://bio-bwa.sourceforge.net). The SHORE consensus analysis was applied to find variant and reference calls in each individual sample. The 192 genotyping data were filtered by selecting markers with at least 80% individuals genotyped and the following settings: 0.3 to 0.7 concordance and minimum 5-read depth to define heterozygous calls. Phenotyping data were assigned to the 192 genotyped F$_2$ individuals. The QTL package of R/qtl software (https://rqtl.org) was employed to perform QTL analysis. RAD-seq data have been deposited to the Dryad repository with the dataset identifier https://doi.org/10.5061/dryad.h70rxwdkx.

**IF1 synthesis and recombinant expression**. In vitro transcription and translation (TnT) were performed with full-length IF1 cloned into pET-pDEST42 (Thermo Fisher Scientific, for primer sequences, see Supplementary Table 2) and the TnT® Quick Coupled Transcription/Translation System (for T7 promoter, Promega) according to the manufacturer's protocol. The provided luciferase construct was used as a control.

For recombinant protein expression in *E. coli* BL21AI, full-length IF1 from different bacteria or IF1 fragments I7–R72 and M1–I67 were cloned into pET-pDEST42 (Thermo Fisher Scientific, for primer sequences, see Supplementary Table 2). Protein expression was induced by 0.2% L-arabinose and 1 mM IPTG for 24 h at 17 °C and 220 rpm. The cell pellets were resuspended in binding buffer A containing 20 mM KP$_i$, pH 7.4, 500 mM KCl and 50 mM imidazole. After sonication and centrifugation (45 min, 20,000 *g*, 4 °C), the clear supernatant was applied to a 1 ml HisTrapFF column (GE Healthcare). C-terminally 6xHis-tagged proteins were collected in 1 ml fractions with an elution-buffer gradient (1 ml/min, 0–100% buffer B: 20 mM KP$_i$, pH 7.4, and 500 mM KCl, 500 mM imidazole). The protein concentration was determined according to Bradford[54] using the Roti-Quant solution (Carl Roth).

For recombinant protein expression in *Pichia pastoris* GS115 (Multi-Copy Pichia Expression Kit Instructions, Thermo Fisher Scientific), constructs encoding IF1, mutant versions of IF1, CspA, and CspA with IF1, the α-helix, and *Arabidopsis* chloroplast-derived IF1 were cloned into the secretory expression plasmid pPICZalphaA. IF1 mutant constructs were generated using the GeneArt Site-Directed Mutagenesis System (Thermo Fisher Scientific) and AccuPrime Pfx DNA Polymerase (Thermo Fisher Scientific). CspA, CspA–IF1 helix, and *Arabidopsis* chloroplast-derived IF1 coding sequences were amplified from synthetic gene constructs (Eurofins). Sequence information on primers and synthetic genes used in this study can be found in Supplementary Table 2. Protein purification from *P. pastoris* culture medium was achieved by affinity chromatography on HisTrap excel columns (Cytiva, Marlborough, MA, USA; equilibrated in 20 mM KP$_i$, pH

7.4, 500 mM KCl). Following washing (20 mM KP$_i$, pH 7.4, 500 mM KCl, and 20 mM imidazole) and elution (buffer gradient 0–500 mM imidazole in equilibration buffer), IF1 or CspA containing fractions were pooled and dialyzed against H$_2$O. Protein concentrations were calculated by UV spectroscopy (wavelength λ280) using extinction-coefficient (ε280) estimates determined using the protparam tool (http://web.expasy.org/protparam). Determinations were verified by SDS-PAGE using a standard protein solution.

**Generation of transgenic plants**. Accession-specific *RLP32* coding sequences with or without the native promoter were amplified with Pfu DNA Polymerase (Thermo Fisher Scientific) using primers listed in Supplementary Table 2, and cloned into the pCR®8/GW/TOPO®-TA vector (Thermo Fisher Scientific). For 35S-promoter-driven expression in *N. benthamiana*, *RLP32* coding sequence was fused to a C-terminal GFP tag in pB7FWG2.0[55]. For native promoter-driven expression in *Arabidopsis*, *RLP32* promoter and coding sequence were recombined into pGWB1 (no tag) or pGWB4 (C-terminal GFP tag)[56], respectively. For stable plant transformations, the *RLP32* allele from accession Col-0 was used. Transient and stable transformations of *Arabidopsis* and *N. benthamiana* were performed as described previously[17].

**Immune assays and bacterial inoculations**. *Arabidopsis* leaves were cut into pieces (2 × 3 mm size) and floated on H$_2$O for 12 h in petri dishes before quantification of ethylene accumulation and the production of reactive oxygen species (ROS). For assaying ethylene production, the elicitor was added into a glass tube containing 200 µl of MES buffer, pH 5.7, with three leaf pieces for 2 h, followed by using gas-chromatography analysis (Shimadzu GC-14). For ROS-burst measurement, two leaf pieces/well of a white 96-well plate were incubated in 100 µl of 20 µM L-012 (a luminol derivative) containing 0.5 µg/ml peroxidase solution. Measurements of luminescence were performed in a 96-well luminometer (Mithras LB 940, Berthold Technologies) for 1 h with signal-integration times of 1 s in 1 min intervals. To visualize callose apposition, leaves were harvested 24 h after infiltration of elicitors. Quantification of callose was performed by counting selected pixels and calculated in % relative to the respective image section of the leaf surface. Pictures were analyzed using the Adobe Photoshop CS6 Magic tool, hereby removing background and leaf veins within a certain color range. For MAPK-activity assays, leaves were harvested 15 min after infiltration with control or elicitor samples and frozen in liquid nitrogen prior to protein extraction in 20 mM Tris-HCl pH 7.5, 150 mM NaCl, 1 mM EDTA, 1% Triton X-100, 0.1% SDS, 5 mM DTT, Complete Protease Inhibitor Mini, EDTA-free (Roche, Mannheim), and PhosStop Phosphatase Inhibitor Cocktail (Roche, Mannheim). After pelleting cell debris (10 min, 16000 *g*, 4 °C), the supernatant (30 µg protein) was separated on 10% SDS-PAGE, transferred to nitrocellulose, and activated MAPK6, 3 and 4 were detected by protein blotting using the rabbit anti-phospho-p44/42-MAPK antibody (Cell Signaling Technology, 9101, 1:3000 dilution) with secondary anti-rabbit antibody coupled to alkaline phosphatase (Sigma-Aldrich, A3687, 1:10,000 dilution). For the histochemical detection of GUS enzyme activity, whole leaves of *pPR-1:GUS* transgenic *Arabidopsis* were placed in 1 ml of 50 mM sodium phosphate, pH 7, 0.5 mM potassium ferrocyanide, 0.5 mM potassium ferricyanide, 10 mM EDTA, pH 8, 0.1% Triton X-100, and 0.5 mg/ml 5-bromo-4-chloro-3-indolyl-ß-D-glucuronide (X-gluc, X-Gluc-DIRECT). After vacuum infiltration, leaves were incubated at 37 °C overnight, and chlorophyll was subsequently removed by several washings in 70% ethanol[17,30]. In *Arabidopsis* plants, a 24 h priming using 1 µM nlp20 or IF1 and subsequent leaf infiltration with a final cell density of 10$^4$ cfu/ml *Pseudomonas syringae* pv. *tomato* DC3000 (*Pst*DC3000) was performed as described[17]. Spray inoculations[57] were performed with 10$^6$ cfu/ml *Pst*DC3000 or PstDC3000COR$^-$, a coronatine-deficient strain carrying mutations in *cfa* and *cma* loci[34]. *N. benthamiana* plants were likewise infected with *Pst*DC3000Δ*hrcC*[35], PstDC3000Δ*hopQ1-1*[37], or *P. syringae* pv. *tabaci*[36] at a final cell density of 2*10$^4$ cfu/ml and harvested after 0 and 4 days.

**Immunoprecipitation assays and in vivo cross-linking**. For co-immunoprecipitation, RLP32-GFP (in pB7FWG2[55]) and co-receptors SOBIR1-HA (pGWB14[56]) and BAK1-myc (pGWB17[56]) were transiently expressed in *N. benthamiana*, and leaf material was harvested either directly or 5 min after infiltration of 1 µM IF1. About 200 mg of ground leaf material was subjected to protein extraction and immunoprecipitation using GFP-Trap beads (ChromoTek, IZB Martinsried, Germany)[17]. In vivo cross-linking experiments were conducted using leaves of *N. benthamiana* stably expressing *p35S::RLP32-GFP* (in pB7FWG2[55]), or transiently expressing *p35S::RLP23-GFP*[17]. In brief, leaves were infiltrated with 30 nM biotinylated IF1, 30 nM biotinylated nlp24, or 10 mM MgCl$_2$, with or without 30 µM unlabeled synthesized IF1 or nlp20 as competitor. Five min after peptide treatment, 2 mM ethylene glycol bis(succinimidyl succinate) (EGS) was infiltrated into the same leaves, and leaf samples were harvested after further 15 min. About 300 mg of the sample was used for protein extraction and immuno-adsorption to GFP-Trap beads as described[17,58]. Membrane proteins were extracted by grinding samples solubilized in cold extraction buffer containing 50 mM Tris-HCl pH 8, 150 mM NaCl, 10% glycerol, 1% (w/v) Nonidet P-40, 0.5% (w/v) sodium deoxycholate, and protease-inhibitor cocktail (Roche, Mannheim). Protein solutions were immunoadsorbed by GFP-Trap beads at 4 °C for 1 h followed by washing beads three times with extraction buffer. Protein blots were probed either directly with a streptavidin–alkaline phosphatase conjugate (Roche, 11089161001, 1:2500 dilution) or with antibodies raised against GFP (Torrey Pines Biolabs, TP401, 1:4000 dilution), HA- (Sigma-Aldrich, H3663, 1:2000 dilution), or Myc tag (Sigma-Aldrich, C3956, 1:5000 dilution) followed by staining with secondary antibodies coupled to alkaline phosphatase (Sigma-Aldrich, A4312, 1:10,000 dilution for HA-tag antibody, A3687, 1:10,000 dilution for GFP- and Myc-tag antibodies) and CDP-Star (Roche, 11759051001) as substrate. Chemiluminescence was detected using a CCD camera (Viber Louromat, PeqLAB).

**Statistical analysis**. Statistical analyses were carried out with JMP (SAS Institute Inc, Cary, North Carolina). All normal-distribution data sets and small data sets ($n < 6$) were evaluated using Dunnett's test with control. Data sets with non-normal distribution were evaluated with nonparametric tests: Mann–Whitney-U-test for comparison of two samples or Steel test with control for multiple-comparison analysis. Exact *P*-values are provided in the figures and the Source data file. EC$_{50}$ values and curve fit were calculated using 4 P Rodbard Model comparison (four-parametric logistic regression).

**Reporting summary**. Further information on research design is available in the Nature Research Reporting Summary linked to this article.

# Data availability
All data are available within this article and its Supplementary Information. RLP32 amino acid sequences from *A. thaliana* accessions were obtained from the 1001 Genomes project (http://signal.salk.edu/atg1001/3.0/gebrowser.php). Original gel blots are shown in the Source data file. RAD-seq data have been deposited to the Dryad repository with the dataset identifier doi:10.5061/dryad.h70rxwdkx. Mass spectrometry proteomics data have been deposited to the ProteomeXchange Consortium via the PRIDE[59] partner repository with the dataset identifier PXD031124. Source data are provided with this paper.

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

## Acknowledgements

Research in the lab of T.N. was funded by DFG grants Nu70/9-1, Nu70/9-2, Nu 70/15-1, Nu70/16-1, Nu70/17-1, and SFB1101. RAD-seq work was supported by the Max Planck Society (to D.W.) and the Academic Research Fund (MOE2019-T2-1-134) from the Ministry of Education, Singapore (to E.C.). We thank Caterina Brancato for assistance in plant transformation, and Mirita Franz-Wachtel and Ilya Bezrukov for processing proteomics and RAD-Seq data.

## Author contributions

L.F., K.F., E.M., I.A., E.C. and T.N. conceived and designed the experiments; L.F. conducted genotyping-by-sequencing-based identification of RLP32; E.M. isolated and characterized IF1; K.F. characterized IF1/RLP32 interactions and conducted, together with A.J. and Y.S., plant-infection assays; I.A., R.N.P., C.H. and L.Z. biochemically characterized IF1 and RLP32; L.F., K.F., E.M., I.A., R.N.P., L.Z., M.A., S-T.K., E.C., D.W., A.A.G., and T.N. analyzed data; and R.N.P., E.C., C.Z., A.A.G. and T.N. wrote the paper. All authors discussed the results and commented on the paper.

## Funding

## Competing interests

The authors declare no competing interests.
