## [Peer Review File · Nature Communications]

Genotyping-by-sequencing-based identification of Arabidopsis pattern recognition receptor RLP32 recognizing proteobacterial translation initiation factor IF1REVIEWER COMMENTS

Reviewer #1 (Remarks to the Author):

In this study, Fan et al identified a translation initiation factor 1 (IF1) derived from bacteria that triggers immune responses in Arabidopsis, Brassica rapa and B. oleracea. By assaying the natural variations within Arabidopsis accessions, the authors identified the receptor-like protein 32 (RLP32) mediates IF1 perception. The work is interesting and will definitely add to the growing knowledge on how microbe-associated molecular patterns could be perceived by plants to initiate plant immunity. However, several issues need to be addressed:

Major issues

- 1, The molecular patterns often trigger a series of immune responses in sensitive plants. For IF1, the authors only tested ethylene throughout the manuscript. Although the authors tested the different immune responses triggered by RsE in the Supplementary Fig. 1, as far as I understand, RsE is a mixture of protein extracts, which could not reflect the elicitor activity of IF1. The authors should evaluate the different immune responses triggered by IF1, and compare the activity of IF1 with RsE.
- 2, The authors claimed that RLP32 binds to IF1 and forms complexes with SOBIR1 and BAK1 in Fig 6. But in the Fig.6b, the N. benthamiana leaves expressing RLP32-GFP, BAK1-myc and SOBIR1-HA were treated with RsE (+), but not by IF1. This assay should be performed using IF1 as elicitor.
- 3, Since this study exploits natural variations within Arabidopsis accessions to identify the IF1 receptor, it would be interesting to check whether the variations in IF1 perception in different Arabidopsis accessions is tightly linked to the polymorphisms of RLP32. This would help to interpret whether the sequence polymorphisms of RLP32 determine the recognition (binding affinity to IF1), or the signal transduction (binding to BAK1 or SOBIR1)? Especially in the Supplementary Fig. 3a, many of the tested accessions produced more ethylene than Col-0 in response to RsE. The RLP23 alleles in these accessions probably elevate the immune responses triggered by IF1, which are more favorable to be deployed to improve plant resistance.

Minors :

1. The authors claimed that all the rlp32 mutants are insensitive to RsE in Fig.1e. However, the mutants show almost half amount of ethylene accumulation when compared with the control Col-0. It is better to include the mock treatment in this assay to support their conclusions.
2. In Fig. 4e, did the authors test additive effect of IF1 and elf18 at a conc. higher than 0.1 nM?
3. In the discussion, the authors claimed that all known PAPMs can be ascribed to small epitopes, with the identified IF1 as an exception. Are there any other receptors that recognize PAMPs in a manner depending on their structures, for example Avr9?

Reviewer #2 (Remarks to the Author):

This manuscript reports the identification and characterization of a novel bacterial elicitor of immunity (EF1) and its associated plant receptor (RLP32), combining thorough biochemical and genetic approaches. On one hand, this work provides additional mechanistic insights on the complex perception of microbial pathogens by plasma membrane-localized immune receptors. On the other, the fact that this receptor seems to be mostly limited to Brassicas suggests that its transfer to other plant families may enhance resistance to bacterial pathogens. The resulting manuscript is very interesting and straightforward, although some interesting findings could provide additional validation or discussion.

1- Contribution of RLP32 to resistance against bacterial pathogens.

If RLP32 indeed contributes to basal immunity against adapted pathogens, it could be expected that *rlp32* mutants display enhanced susceptibility (in practice, this depends a lot on the combination of PRR mutant, pathogen strain, and experimental conditions). The title in Figure 5 states that RLP32 mediates bacterial resistance, and shows that EF1 induces resistance against Pst DC3000, which is lost in *rlp32*, but the design/representation of this experiment does not allow a direct comparison between Col-0 and *rlp32* mutants. A targeted experiment/representation to provide a direct comparison would help to determine this point. However, looking at the graphs, it does not seem that the lack of *rlp32* enhances basal susceptibility to Pst DC3000, although probably the tested strain or inoculation method is not ideal for this purpose. In this regard, the inoculation method used in Figure 5 (infiltration or spray) should be indicated in the figure legend and the methods. Then, to address this point, the authors could test the potential susceptibility of the *rlp32* mutants, compared to Col-0, after inoculation with different Pst strains: if Pst DC3000 is too virulent to detect differences, hypo-virulent strains, such as Pst COR- or T3E mutants could be used, and probably spray inoculation would expand the dynamic range of detection in such assay.

2- Potential of RLP32 to confer resistance to bacterial pathogens in other plant families.

Figure 5d shows a very interesting and promising result in this regard, but the choice of bacterial strain for this assay is not ideal, since it does not show whether RLP32 expression enhances resistance to pathogenic bacteria in *N. benthamiana* (since the Pst *hrcC* mutant is non-pathogenic, as it can be shown by the very weak growth in this assay). Inoculation with pathogenic strains, such as Pst *hopQ1* mutant or *P.s. tabaci*, could strengthen this key point. Please also note that the label of the y axis shows Pst DC3000 instead of the *hrcC* mutant strain used.

3- IF1 conservation among bacteria.

It seems that IF1 is conserved among proteobacteria. Considering a potential use of RLP32 to enhance disease resistance, I wonder if IF1 sequence is also conserved in beneficial bacteria, and whether RLP32 would mediate resistance if perceiving also these IF1s. Some discussion about this could be provided. Also, do chloroplasts contain any protein similar to IF1 that could be involved in translation in chloroplasts? (and could this protein be perceived by RLP32 as a damage signal?)

4- Different sensitivity of Arabidopsis accessions to Rse/EF1.

I find really interesting that different Arabidopsis accessions display very different sensitiveness to Rse, which brings about several questions:

- Is this due to differences in RLP32 itself? This seems difficult to predict based on sequence. If this is due to RLP32 itself, this should be possible to compare upon transient expression of the different versions in *N. benthamiana*. On the other hand, if this is due to downstream signaling, these accessions should also show different responsiveness to other PAMPs.

- It is not indicated in the manuscript text (nor in the methods) whether the RLP32 expression/complementation constructs use the version from Col-0 (which I guess is expected, but not totally clear).

- If RLP32 from hyper-responsive accessions also confers hyper-responsiveness upon transient expression in *N. benthamiana* (compared with the Col-0 version, if that is the one used in this work), it would be interesting to use such version(s) to test resistance against pathogens in the previous point 2.

5- RLP32 protein complex.

It is interesting that RLP32 shows constitutive association with SOBIR1 and ligand-dependent association with BAK1. However, the result shown in figure 6b does not seem to be properly validated. First, the IP control should contain some GFP-tagged protein (or GFP itself), since, at the moment, a lack of protein in that negative control does not exclude the association between tags in the experimental conditions used. Second, the current figure shows cropped blots, where the bands in the experiment and the control lanes are separated. It would be important to show a continuous image showing the control lane and the lanes containing the tested proteins in the same exposure, since, at the moment, it is not possible to see if there would be similar bands in the control lane in the conditions/exposure used. Please also note that there seems to be a GFP band (between 130-180 KDa in the input of the control that is not supposed to contain any GFP-tagged protein); is this an unspecific band?

6- Additivity between perception of EF1 and other PAMPs.

Figure 4e shows that application of IF1 and Ef-Tu does not result in additive or synergistic increases in defense outputs, and the discussion mentions that this result suggests that individual PRR systems may act rather independently. To me, this sounds like downstream signaling is common between both PRRs,

and maybe this sentence in the discussion could be rephrased. Actually, I think it would also be interesting to test if PAMPs from different nature (besides the somehow related EF1 and Ef-Tu) act additively, but I don't think such analysis is required in this manuscript.

Other comments:

- Figure S1: it would be informative to indicate in the figure legend what PEN is, in order to be relevant as a control (otherwise it needs to be looked at in the methods).

- Several accessions seem to show stronger responsiveness than ICE135. Why was it chosen as hypersensitive accession for the identification of the receptor?

- The manuscript text mentions "pathogen infection" in several instances where "inoculation" should have been used, since "infection" is associated to pathogen proliferation and the development of disease. Clear examples are the use of "days post infection" or the use of "infection" to refer to the inoculation with a non-pathogenic strain shown in Figure 5d, but I would recommend to check the whole text for accuracy in this regard.

Alberto Macho

Reviewer #3 (Remarks to the Author):

This manuscript demonstrates identification of Arabidopsis pattern recognition receptor RLP32 recognizing proteobacterial translation initiation factor IF1. It has been previously reported that *Ralstonia solanacearum* produces Arabidopsis defense elicitors other than bacterial flagellin. Authors used protein fractions (RsE) from liquid culture-grown *R. solanacearum* elicited plant defense response in the Arabidopsis *fls2* and *efr* mutant. Then, authors obtained three RsE insensitive accessions showing reduced ethylene production from 106 Arabidopsis accessions. To identify the gene (RLP32) controlling the sensitivity, authors used a restriction site-associated DNA sequencing (RAD-seq) method and a population segregating the RsE sensitivity. Genotyping-by-sequencing and mutant analysis identified the LRR type RLP32 among 6 candidates as the gene conferring RsE sensitivity. Authors obtained used

protein extracts from non-pathogenic plant *E. coli* and found that protein translation initiation factor 1 (IF1) induced RLP32-dependent ethylene production. They showed that IF1 is a wide spread bacterial pattern and that *R. solanacearum* IF1 accounts for the elicitor activity of in RsE. They showed that IF1 elicitor activity required tertiary fold features for IF1 receptor activation, which is different from other immunogenic patterns that are typically short, conserved peptide fragments. Transgenic expression of RLP32 conferred IF1 sensitivity to IF1-insensitive *Arabidopsis* accessions (*rlp32* mutants) and IF1-insensitive *Nicotiana benthamiana*. Authors demonstrated that RLP32 confers resistance to *Pseudomonas syringae* infection which is similar to previously identified PRRs. Furthermore, authors showed that RLP32 specifically bound IF1 and formed complexes with LRR receptor kinases SOBIR1 and BAK1 to mediate signaling. Finally authors showed that IF1 sensitivity is restricted to some related Brassicaceae species.

Overall, the paper is well written and provides strong evidences that IF1 is a novel bacterial pattern and RLP32 is a novel PRR. It would be nicer if authors could include a paragraph describing evolution of RLP32. Why some accessions retain the function of RLP32 and some are not?

Minor point: why ethylene production in mock treatments of Brassica is so high? (Fig 6d)

Tübingen, November 30, 2021

Here we provide point-by-point replies to all Referee queries and explanations for changes we have made to the original MS. To facilitate assessment, we have copied reviewer's queries (Q) ahead of our replies (R).

Reply to comments of Reviewer 1

Major issues

Q1: The molecular patterns often trigger a series of immune responses in sensitive plants. For IF1, the authors only tested ethylene throughout the manuscript. Although the authors tested the different immune responses triggered by RsE in the Supplementary Fig. 1, as far as I understand, RsE is a mixture of protein extracts, which could not reflect the elicitor activity of IF1. The authors should evaluate the different immune responses triggered by IF1, and compare the activity of IF1 with RsE.

R1: We thank Referee 1 for her/his valuable recommendation and agree that the requested information is very important. We have now shown that IF1 triggers ROS production, transcript accumulation of defense marker genes, callose apposition and MAPK activation in an RLP32-dependent manner (new Supplementary Fig. 12). These findings together with IF1-induced production of the plant stress hormone ethylene (Fig. 2) and IF1-induced priming of immunity (Fig. 5a-c) demonstrate that IF1 triggers a comprehensive defense response in *Arabidopsis* that is comparable to that triggered by RsE (Supplementary Fig. 1).

Q2: The authors claimed that RLP32 binds to IF1 and forms complexes with SOBIR1 and BAK1 in Fig 6. But in the Fig.6b, the *N. benthamiana* leaves expressing RLP32-GFP, BAK1-myc and SOBIR1-HA were treated with RsE (+), but not by IF1. This assays should be performed using IF1 as elicitor.

R2: We again agree with Reviewer 1 and have added this information in Fig. 6b.

Q3: Since this study exploits natural variations within *Arabidopsis* accessions to identify the IF1 receptor, it would be interesting to check whether the variations in IF1 perception in different *Arabidopsis* accessions is tightly linked to the polymorphisms of RLP32. This would help to interpret whether the sequence polymorphisms of RLP32 determine the recognition (binding affinity to IF1), or the signal transduction (binding to BAK1 or SOBIR1)? Especially in the Supplementary Fig. 3a, many of the tested accessions produced more ethylene than Col-0 in response to RsE. The RLP32 alleles in these accessions probably elevate the immune responses triggered by IF1, which are more favorable to be deployed to improve plant resistance.

R3: Reviewer 1 raises the intriguing question whether inspecting sequence polymorphisms in *RLP32* alleles in various accessions would unveil residues required for RLP32 function. We have retrieved publicly available RLP32 sequence information for 93 of the 106 *A. thaliana* accessions initially screened for insensitivity to RsE (Supplementary Fig. 9). Such sequences included those from 7 RsE-insensitive and from 19 RsE-hypersensitive

accessions. In 13 accessions, sequence ambiguities hampered identification of RLP32-encoding sequences. Substantial polymorphisms among RLP32 orthologs were found, which made predictions about sites causal for RsE- sensitivity rather difficult. For example, RLP32 sequences from RsE-insensitive accession ICE21 and RsE-sensitive ICE71 differed in only three amino acids (ICE71 residues S204, V764, S809). Likewise, RLP32 (ICE21) and RsE-sensitive RLP32 (Col-0) sequences differ by only four residues (Q116, D613, E764, G837). However, polymorphisms observed in RLP32 (ICE21) are not conserved among RsE-insensitive accessions, and polymorphisms in RLP32 from Col-0 and ICE71 are not conserved in RsE sensitive accessions (Supplementary Fig. 9). Because of these findings and because *RLP32* alleles from RsE-insensitive accessions are only distantly related (Supplementary Fig. 10), we conclude that loss of canonical RLP32 activity is determined by several unrelated sequence polymorphisms and may have occurred repeatedly during evolution. We have included this new information which reads as follows:

Inspection of RLP32 protein sequences from 93 accessions including those from 7 RsE-insensitive and 19 RsE-hypersensitive accessions (relative to RLP32 activity observed in Col-0) (Supplementary Fig. 3) revealed rather diverse polymorphisms among RLP32 orthologs (Supplementary Fig. 9), thus making predictions about mutations causal for loss of RsE-sensitivity difficult. For example, RLP32 sequences from RsE-insensitive accession ICE21 and RsE-sensitive ICE71 differed in only three amino acids (ICE71 residues S204, V764, S809). Likewise, RLP32 (ICE21) and RsE-sensitive RLP32 (Col-0) sequences differ by only four residues (Q116, D613, E764, G837). However, polymorphisms observed in RLP32 (ICE21) are not conserved among RsE-insensitive accessions, and polymorphisms in RLP32 from Col-0 and ICE71 are not conserved in RsE sensitive accessions (Supplementary Fig. 9). Because of these findings and because *RLP32* alleles from RsE-insensitive accessions are only distantly related (Supplementary Fig. 10), we conclude that loss of canonical RLP32 activity is determined by several unrelated sequence polymorphisms and may have occurred repeatedly during evolution.

Reviewer 1 further addresses whether accessions that are hypersensitive to RsE and produce more ethylene than Col-0 do so because of more efficient IF1 sensing. Indeed, substantial differences in RsE sensitivities were observed among the 106 *Arabidopsis* accessions tested in our initial screen (Supplementary Fig. 3). Some accessions exhibited lower levels of RsE-induced ethylene production relative to that in accession Col-0, whereas the majority of the accessions tested showed higher ethylene production. RsE-hypersensitive accessions are also more sensitive to IF1 than accession Col-0 (Supplementary Fig. 18a). However, when treated with flg22 or nlp20 patterns, RsE/IF1-hypersensitive accessions also showed increased levels of ethylene production relative to those observed in elicitor-treated Col-0 (Supplementary Fig. 18a). These findings suggest that RsE/IF1 hypersensitivity is not due to accession-specific RLP32 sequence information, but may be due to accession-specific differences in signaling events downstream of receptor activation. In agreement with this notion, transient expression of *RLP32* alleles from Col-0 and IF1-hypersensitive accessions ICE93 and ICE71 (Supplementary Fig. 3) in *N. benthamiana* resulted in similar levels of IF1-induced ethylene production (Supplementary Fig. 18b).

We have included this new information which reads as follows:

Accession-specific differences in IF1 responsiveness are not linked to RLP32 protein sequence polymorphisms. Substantial differences in RsE sensitivities were observed among the 106 *Arabidopsis* accessions tested in our initial screen (Supplementary Fig. 3). Some accessions exhibited lower levels of RsE-induced ethylene production relative to that in accession Col-0, whereas the majority of the accessions tested showed higher ethylene production. RsE-hypersensitive accessions are also more sensitive to IF1 than accession Col-0 (Supplementary Fig. 18a). However, when treated with flg22 or nlp20 patterns, RsE/IF1-hypersensitive accessions also showed increased levels of ethylene production relative to those observed in elicitor-treated Col-0 (Supplementary Fig. 18a). These findings suggest that RsE/IF1 hypersensitivity is not due to accession-specific RLP32 sequence information. In agreement with this notion, transient expression of *RLP32* alleles from Col-0 and IF1-hypersensitive accessions ICE93 and ICE71 (Supplementary Fig. 3) in *N. benthamiana* resulted in similar levels of IF1-induced ethylene production (Supplementary Fig. 18b).

Minor issues

Q4: The authors claimed that all the *rlp32* mutants are insensitive to RsE in Fig. 1e. However, the mutants show almost half amount of ethylene accumulation when compared with the control Col-0. It is better to include the mock treatment in this assay to support their conclusions.

R4: In Fig. 1a, we compare ethylene levels in mock- vs. RsE-treated *Arabidopsis* accessions. RsE-insensitive accessions exhibit some ethylene production that is higher than that observed in mock-treated plants. This is likely due to elicitor activity in partially purified RsE that is unrelated to IF1. In Fig. 2a, c-d, we have used pure IF1 to elicit ethylene production in *rlp32* mutants and show that these levels are very similar to those evoked by mock treatment. This supports the notion that RsE contains residual elicitor activity unrelated to IF1. In Fig. 1e, we compare RsE-induced ethylene production in various *Arabidopsis* mutant genotypes and accessions. This

figure aims at comparing ethylene levels in RsE-treated *rlp32* mutants and RsE-insensitive accessions ICE21 and ICE73, which are similar. We agree with Referee 1 that negative controls are mandatory for biological experiments. We think, however, that ICE73 serves as appropriate negative control in the experiment shown in Fig. 1e.

Q5: In Fig. 4e, did the authors test additive effect of IF1 and elf18 at a conc. higher than 0.1 nM?

R5: IF1 and EF-Tu are distinct components of the proteobacterial protein translation machinery that are perceived by *Arabidopsis* PRRs RLP32 and EFR, respectively. We have tested whether simultaneous action of both patterns, a scenario likely occurring during bacterial infection, has additive or synergistic effects on plant defense activation. IF1 and EF-Tu fragment elf18 were applied either alone or in combination at concentrations below or corresponding to the EC₅₀ values of the respective patterns (Fig. 2b). These concentrations were chosen because additive or synergistic effects on defense activation are meaningful only at rather low concentrations. At elicitor concentrations above the respective EC₅₀ values plant responses are already saturated upon treatment with a single stimulus. Thus, the use of higher elicitor concentrations was not pursued here.

Q6: In the discussion, the authors claimed that all known PAPMs can be ascribed to small epitopes, with the identified IF1 as an exception. Are there any other receptors that recognize PAMPs in a manner depending on their structures, for example Avr9?

R6: To our knowledge, IF1 is the first PAMP whose tertiary structure fold is shown to be required for immune receptor-mediated activation of plant pattern-triggered immunity (PTI). We agree with Referee 1 that the tertiary fold of Avr9 has been shown to be required for elicitation of Cf-9-dependent ETI in tomato. However, evidence for binding of Avr9 to the LRR ectodomain of Cf-9 is lacking. Indeed, concerted action of many laboratories has failed to demonstrate such interaction by various experimental approaches (R. Luderer *et al.*, *Mol-Plant-Microbe Interact.*, 2001; No evidence for binding between resistance gene product Cf-9 of tomato and avirulence gene product AVR9 of *Cladosporium fulvum*). Likewise, a high-affinity binding site for Avr9 has been reported from tomato cultivars lacking or possessing Cf-9 (M. Kooman-Gersmann *et al.* *Plant Cell*, 1996; A high-affinity binding site for the Avr9 peptide elicitor of *Cladosporium fulvum* is present on plasma membranes of tomato and other solanaceous plants), thus making it unlikely that Cf-9 directly binds Avr9. Hence, detailed information about the recognition of Avr9 and its primary receptor is still lacking.

Reply to comments of Reviewer 2

Q1: Contribution of RLP32 to resistance against bacterial pathogens. If RLP32 indeed contributes to basal immunity against adapted pathogens, it could be expected that *rlp32* mutants display enhanced susceptibility (in practice, this depends a lot on the combination of PRR mutant, pathogen strain, and experimental conditions). The title in Figure 5 states that RLP32 mediates bacterial resistance, and shows that EF1 induces resistance against Pst DC3000, which is lost in *rlp32*, but the design/representation of this experiment does not allow a direct comparison between Col-0 and *rlp32* mutants. A targeted experiment/representation to provide a direct comparison would help to determine this point. However, looking at the graphs, it does not seem that the lack of *rlp32* enhances basal susceptibility to Pst DC3000, although probably the tested strain or inoculation method is not ideal for this purpose. In this regard, the inoculation method used in Figure 5 (infiltration or spray) should be indicated in the figure legend and the methods. Then, to address this point, the authors could test the potential susceptibility of the *rlp32* mutants, compared to Col-0, after inoculation with different Pst strains: if Pst DC3000 is too virulent to detect differences, hypo-virulent strains, such as Pst COR- or T3E mutants could be used, and probably spray inoculation would expand the dynamic range of detection in such assay.

R1: We fully agree with Reviewer 2 who suggested to conduct additional bacterial growth assays in order to elucidate the role of RLP32 in plant basal immunity. We agree that lack of RLP32 does not increase plant susceptibility to PstDC3000 relative to Col-0 when bacteria are infiltrated into leaves (Fig. 5). The method of bacterial application is now referred to in the legend of Fig. 5 as requested. In addition, we have conducted bacterial spray inoculation experiments in Col-0 and *rlp32* genotypes (see new Supplementary Fig. 16a, b). However, application of neither PstDC3000 nor PstDC3000COR strains gave rise to higher bacterial titers. We thus conclude that RLP32 may not significantly contribute to bacterial resistance under our plant growth and experimental conditions. As noted below (and in the discussion), we believe that this may be due to complex arrays of *Arabidopsis* PRRs that recognize structurally unrelated patterns from PstDC3000 and related strains.

Identification of the ligand-receptor pair IF1/RLP32 illustrates that recognition of a single pathogen species by a given host plant can be enormously complex. In addition to RLP32, *Arabidopsis* has evolved at least five additional receptor systems to sense the bacterial pathogen *P. syringae*. These receptors comprise well-studied FLS2 and EFR as well as sensors for bacterial mc-3-OH-fatty acids (LORE), peptidoglycan (LYM1-LYM3-CERK1) and xanthine/uracil permease (XPS1)^{16,18,19,23,24}. We can only speculate why a single plant species

may have evolved such a number of redundant microbial sensing systems. It seems to be reasonable to assume that plant PRR complexity may not be brought about by co-evolution of these two organisms alone, but may result from multiple independent evolutionary processes that were driven by exposure to various microbial threats. Redundancy in *Arabidopsis* PRRs may explain, however, accession-specific losses of recognition specificities that have been reported for most microbial sensor systems in this plant.

Q2: Potential of RLP32 to confer resistance to bacterial pathogens in other plant families. Figure 5d shows a very interesting and promising result in this regard, but the choice of bacterial strain for this assay is not ideal, since it does not show whether RLP32 expression enhances resistance to pathogenic bacteria in *N. benthamiana* (since the Pst hrcC mutant is non-pathogenic, as it can be shown by the very weak growth in this assay). Inoculation with pathogenic strains, such as Pst hopQ1 mutant or *P. s. tabaci*, could strengthen this key point. Please also note that the label of the y axis shows Pst DC3000 instead of the hrcC mutant strain used.

R2: Again, we thank Reviewer 2 for his valuable suggestion that helped us to improve the significance of our findings. As suggested, we have infiltrated *N. benthamiana* plants stably transformed with *AtRLP32* with *P. syringae* pv. *tabaci* or *PstDC3000ΔhopQ1-1* strains, respectively. As shown in new Fig. 5e, *Ps. tabaci* bacterial titers were significantly lower in RLP32-transgenic plants relative to control plants, suggesting that RLP32 confers partial resistance to this tobacco pathogen. In contrast, no such effect was observed when strain *PstDC3000ΔhopQ1-1* was used (new Supplementary Fig. 16c). Altogether, use of additional strains in plant infection models helped us to gain more comprehensive information on how RLP32 contributes to plant immunity.

Q3: IF1 conservation among bacteria. It seems that IF1 is conserved among proteobacteria. Considering a potential use of RLP32 to enhance disease resistance, I wonder if IF1 sequence is also conserved in beneficial bacteria, and whether RLP32 would mediate resistance if perceiving also these IF1s. Some discussion about this could be provided. Also, do chloroplasts contain any protein similar to IF1 that could be involved in translation in chloroplasts? (and could this protein be perceived by RLP32 as a damage signal?)

R3: IF1 sequences are found throughout proteobacterial species including plant non-pathogenic (beneficial) species. In addition, sequences of plant chloroplast-derived IF1 are similar to those of elicitor-active microbial IF1 (new Supplementary Fig. 14f). We have now tested whether IF1 proteins from *Lysobacter* spp. or *Rhizobacter* spp. (present in the *Arabidopsis* microbiome, Y. Bai *et al. Nature*, 2015; Functional overlap of the *Arabidopsis* leaf and root microbiota) possess elicitor activity in *Arabidopsis*. As shown in new Supplementary Fig. 14b, both proteins harbor residual elicitor activities, which are, however, approximately two orders of magnitude lower than that of *E. coli* IF1. *Arabidopsis* chloroplast-derived IF1 did not exhibit elicitor activity at all when applied to *Arabidopsis* itself or to *N. benthamiana* plants transiently expressing RLP32 (new Supplementary Fig. 14 c-f). We refer to these new findings in the Results section as follows:

Synthetic IF1 based on genome sequences from *Lysobacter* or *Rhizobacter* strains that are associated with the *Arabidopsis* root microbiome had substantially less elicitor activity than *E. coli* IF1 (Supplementary Fig. 14b, f). Notably, *Arabidopsis* chloroplast-derived IF1 neither exhibited elicitor activity in *Arabidopsis* itself nor in *N. benthamiana* plants transiently expressing RLP32 (Supplementary Fig. 14c-f), consistent with the absence of autoimmunity triggered by endogenous IF1.

Q4: Different sensitivity of *Arabidopsis* accessions to RsE/EF1. I find really interesting that different *Arabidopsis* accessions display very different sensitiveness to RsE, which brings about several questions:

- Is this due to differences in RLP32 itself? This seems difficult to predict based on sequence. If this is due to RLP32 itself, this should be possible to compare upon transient expression of the different versions in *N. benthamiana*. On the other hand, if this is due to downstream signaling, these accessions should also show different responsiveness to other PAMPs.
- It is not indicated in the manuscript text (nor in the methods) whether the RLP32 expression/complementation constructs use the version from Col-0 (which I guess is expected, but not totally clear).
- If RLP32 from hyper-responsive accessions also confers hyper-responsiveness upon transient expression in *N. benthamiana* (compared with the Col-0 version, if that is the one used in this work), it would be interesting to use such version(s) to test resistance against pathogens in the previous point 2.

R4: Reviewer 2 raises an interesting point that was addressed already by Reviewer 1 (Q3). We therefore add our reply to this query again.

Reviewer 1 (and 2) address whether accessions that are hypersensitive to RsE and produce more ethylene than Col-0 do so because of more efficient IF1 sensing. Indeed, substantial differences in RsE sensitivities were observed among the 106 *Arabidopsis* accessions tested in our initial screen (Supplementary Fig. 3). Some accessions exhibited lower levels of RsE-induced ethylene production relative to that in accession Col-0, whereas the majority of the accessions tested showed higher ethylene production. RsE-hypersensitive accessions are also more sensitive to IF1 than accession Col-0 (Supplementary Fig. 18a). However, when treated with flg22 or nlp20 patterns, RsE/IF1-hypersensitive accessions also showed increased levels of ethylene production relative to those observed in elicitor-treated Col-0 (Supplementary Fig. 18a). These findings

suggest that RsE/IF1 hypersensitivity is not due to accession-specific RLP32 sequence information, but may be due to accession-specific differences in signaling events downstream of receptor activation. In agreement with this notion, transient expression of *RLP32* alleles from Col-0 and IF1-hypersensitive accessions ICE93 and ICE71 (Supplementary Fig. 3) in *N. benthamiana* resulted in similar levels of IF1-induced ethylene production (Supplementary Fig. 18b). This finding also invalidates the use of *RLP32* alleles from RsE/IF1 hypersensitive *Arabidopsis* accessions to engineer bacterial resistance in transgenic plants. We have now also included in the methods section that Col-0 RLP32 was used for stable plant transformations.

We have included this new information which reads as follows:

Accession-specific differences in IF1 responsiveness are not linked to RLP32 protein sequence polymorphisms. Substantial differences in RsE sensitivities were observed among the 106 *Arabidopsis* accessions tested in our initial screen (Supplementary Fig. 3). Some accessions exhibited lower levels of RsE-induced ethylene production relative to that in accession Col-0, whereas the majority of the accessions tested showed higher ethylene production. RsE-hypersensitive accessions are also more sensitive to IF1 than accession Col-0 (Supplementary Fig. 18a). However, when treated with flg22 or nlp20 patterns, RsE/IF1-hypersensitive accessions also showed increased levels of ethylene production relative to those observed in elicitor-treated Col-0 (Supplementary Fig. 18a). These findings suggest that RsE/IF1 hypersensitivity is not due to accession-specific RLP32 sequence information. In agreement with this notion, transient expression of *RLP32* alleles from Col-0 and IF1-hypersensitive accessions ICE93 and ICE71 (Supplementary Fig. 3) in *N. benthamiana* resulted in similar levels of IF1-induced ethylene production (Supplementary Fig. 18b).

Q5: RLP32 protein complex. It is interesting that RLP32 shows constitutive association with SOBIR1 and ligand-dependent association with BAK1. However, the result shown in figure 6b does not seem to be properly validated. First, the IP control should contain some GFP-tagged protein (or GFP itself), since, at the moment, a lack of protein in that negative control does not exclude the association between tags in the experimental conditions used. Second, the current figure shows cropped blots, where the bands in the experiment and the control lanes are separated. It would be important to show a continuous image showing the control lane and the lanes containing the tested proteins in the same exposure, since, at the moment, it is not possible to see if there would be similar bands in the control lane in the conditions/exposure used. Please also note that there seems to be a GFP band (between 130-180 KDa in the input of the control that is not supposed to contain any GFP-tagged protein); is this an unspecific band?

R5: We apologize for the poor quality of the previous Fig. 6b. This is in line with another comment of Reviewer 1 (Q2) who asked us to conduct the experiment shown in Fig. 6b with IF1 instead of RsE. We have replaced the old Fig. 6b with a new version showing the requested experimental data. Uncropped gel images are provided in the Source Data file.

Q6: Additivity between perception of EF1 and other PAMPs. Figure 4e shows that application of IF1 and Ef-Tu does not result in additive or synergistic increases in defense outputs, and the discussion mentions that this result suggests that individual PRR systems may act rather independently. To me, this sounds like downstream signaling is common between both PRRs, and maybe this sentence in the discussion could be rephrased. Actually, I think it would also be interesting to test if PAMPs from different nature (besides the somehow related EF1 and Ef-Tu) act additively, but I don't think such analysis is required in this manuscript.

R6: We apologize for the confusion our previous statement about synergistic patterns has caused. We have rephrased this paragraph as follows:

Bacterial IF1 and translation elongation factor EF-Tu constitute not only different parts of the same molecular machineries, but share similar immunogenic activities in *Arabidopsis* (Fig. 4e). During plant infection, both patterns are likely to trigger immune defenses simultaneously. However, concomitant application of both patterns at non-saturating concentrations did not result in additive or synergistic increases in plant defense output (Fig. 4e). Hence, plants may integrate various external stimuli to mount an appropriate output response, suggesting that signal input may not necessarily be directly linked to plant defense outputs.

We also share with Reviewer 2 his interest about potential additive or synergistic effects of unrelated immunogenic patterns. We also agree, however, that such analyses would be beyond the scope of this study which aimed at elucidating such effects of particularly two patterns that are likely to be released simultaneously during natural infections.

Other comments:

Q7: Figure S1: it would be informative to indicate in the figure legend what PEN is, in order to be relevant as a control (otherwise it needs to be looked at in the methods).

R7: We have revisited the information content of this positive control, which is incremental. We therefore omitted it from previous Supplementary Figures 1 and 2.

Q8: Several accessions seem to show stronger responsiveness than ICE135. Why was it chosen as hypersensitive accession for the identification of the receptor?

R8: This particular hypersensitive accession was chosen for crosses with insensitive accessions ICE21 or ICE73, respectively (Supplementary Fig. 4). Flowering times of these accessions were very similar, which facilitated crossings and F1 seed production.

Q9: The manuscript text mentions "pathogen infection" in several instances where "inoculation" should have been used, since "infection" is associated to pathogen proliferation and the development of disease. Clear examples are the use of "days post infection" or the use of "infection" to refer to the inoculation with a non-pathogenic strain shown in Figure 5d, but I would recommend to check the whole text for accuracy in this regard.

R9: The inappropriate use of the term 'infection' was corrected throughout the text.

Reply to comments of Reviewer 3

Q1: Overall, the paper is well written and provides strong evidences that IF1 is a novel bacterial pattern and RLP32 is a novel PRR. It would be nicer if authors could include a paragraph describing evolution of RLP32. Why some accessions retain the function of RLP32 and some are not?

R1: Reviewer 3 (and 1) raise the intriguing question whether inspecting sequence polymorphisms in *RLP32* alleles in various accessions would unveil residues required for RLP32 function. We have retrieved publicly available RLP32 sequence information for 93 of the 106 *A. thaliana* accessions initially screened for insensitivity to RsE (Supplementary Fig. 9). Such sequences included those from 7 RsE-insensitive and from 19 RsE-hypersensitive accessions. In 13 accessions, sequence ambiguities hampered identification of RLP32-encoding sequences. Substantial polymorphisms among RLP32 orthologs were found, which made predictions about sites causal for RsE- sensitivity rather difficult. For example, RLP32 sequences from RsE-insensitive accession ICE21 and RsE-sensitive ICE71 differed in only three amino acids (ICE71 residues S204, V764, S809). Likewise, RLP32 (ICE21) and RsE-sensitive RLP32 (Col-0) sequences differ by only four residues (Q116, D613, E764, G837).

However, polymorphisms observed in RLP32 (ICE21) are not conserved among RsE-insensitive accessions, and polymorphisms in RLP32 from Col-0 and ICE71 are not conserved in RsE sensitive accessions (Supplementary Fig. 9). Because of these findings and because *RLP32* alleles from RsE-insensitive accessions are only distantly related (Supplementary Fig. 10), we conclude that loss of canonical RLP32 activity is determined by several unrelated sequence polymorphisms and may have occurred repeatedly during evolution. We have included this new information which reads as follows:

Inspection of RLP32 protein sequences from 93 accessions including those from 7 RsE-insensitive and 19 RsE-hypersensitive accessions (relative to RLP32 activity observed in Col-0) (Supplementary Fig. 3) revealed rather diverse polymorphisms among RLP32 orthologs (Supplementary Fig. 9), thus making predictions about mutations causal for loss of RsE-sensitivity difficult. For example, RLP32 sequences from RsE-insensitive accession ICE21 and RsE-sensitive ICE71 differed in only three amino acids (ICE71 residues S204, V764, S809). Likewise, RLP32 (ICE21) and RsE-sensitive RLP32 (Col-0) sequences differ by only four residues (Q116, D613, E764, G837). However, polymorphisms observed in RLP32 (ICE21) are not conserved among RsE-insensitive accessions, and polymorphisms in RLP32 from Col-0 and ICE71 are not conserved in RsE sensitive accessions (Supplementary Fig. 9). Because of these findings and because *RLP32* alleles from RsE-insensitive accessions are only distantly related (Supplementary Fig. 10), we conclude that loss of canonical RLP32 activity is determined by several unrelated sequence polymorphisms and may have occurred repeatedly during evolution.

Q2: Minor point: why ethylene production in mock treatments of Brassica is so high? (Fig 6d)

R2: We have observed that mock (wound)-induced ethylene production in Brassica correlates with leaf age. However, we are unable to provide causal evidence for the observed phenomenon.

We thank you for considering our manuscript for publication.

Yours sincerely,

Dr. Thorsten Nürnberger
(on behalf of all co-authors)

UNIVERSITÄT TüBINGEN
ZMBP Pflanzenbiochemie
Prof. Dr. Thorsten Nürnberger
Auf der Morgenstelle 5
D-72076 Tübingen

REVIEWERS' COMMENTS

Reviewer #1 (Remarks to the Author):

The authors improved their manuscript according to the reviewers suggestions and I am satisfied with the revision of the manuscript. The current manuscript can now be accepted.

Reviewer #2 (Remarks to the Author):

The authors have addressed all my concerns and have made a significant effort to satisfy the requests made by all reviewers. I do not have additional comments

Reviewer #3 (Remarks to the Author):

The authors adequately addressed all the comments addressed by this referee.